# Recessive mutations in muscle-specific isoforms of *FXR1* cause congenital multi-minicore myopathy

María Cristina Estañ[1,2], Elisa Fernández-Núñez[1], Maha S. Zaki[3], María Isabel Esteban[4], Sandra Donkervoort[5], Cynthia Hawkins[6], José A. Caparros-Martin[1,2,17], Dimah Saade[5], Ying Hu[5], Véronique Bolduc[5], Katherine Ru-Yui Chao[7], Julián Nevado[8], Ana Lamuedra[9], Raquel Largo [9], Gabriel Herrero-Beaumont [9], Javier Regadera[10], Concepción Hernandez-Chico[2,11], Eduardo F. Tizzano[2,12], Victor Martinez-Glez [2,8], Jaime J. Carvajal[13], Ruiting Zong[14], David L. Nelson[14], Ghada A. Otaify[3], Samia Temtamy[3], Mona Aglan[3], Mahmoud Issa [3], Carsten G. Bönnemann[5], Pablo Lapunzina[2,8], Grace Yoon[15,16] & Victor L. Ruiz-Perez[1,2,8]

*FXR1* is an alternatively spliced gene that encodes RNA binding proteins (FXR1P) involved in muscle development. In contrast to other tissues, cardiac and skeletal muscle express two FXR1P isoforms that incorporate an additional exon-15. We report that recessive mutations in this particular exon of *FXR1* cause congenital multi-minicore myopathy in humans and mice. Additionally, we show that while *Myf5*-dependent depletion of all FXR1P isoforms is neonatal lethal, mice carrying mutations in exon-15 display non-lethal myopathies which vary in severity depending on the specific effect of each mutation on the protein.

[1] Instituto de Investigaciones Biomédicas "Alberto Sols", CSIC-UAM, 28029 Madrid, Spain. [2] CIBER de Enfermedades Raras (CIBERER), ISCIII, 28029 Madrid, Spain. [3] Department of Clinical Genetics, Human Genetics and Genome Research Division, Centre of Excellence of Human Genetics, National Research Centre, Cairo 12311, Egypt. [4] Departamento de Anatomía Patológica, Hospital Universitario La Paz-IdiPaz-UAM, 28046 Madrid, Spain. [5] Neuromuscular and Neurogenetic Disorders of Childhood Section, National Institute of Neurological Disorders and Stroke, National Institutes of Health, Bethesda, MD 20814, USA. [6] Division of Pathology, Department of Paediatric Laboratory Medicine, The Hospital for Sick Children, University of Toronto, Toronto, ON M5G 1X8, Canada. [7] Center for Mendelian Genomics, Program in Medical and Population Genetics, Broad Institute of MIT and Harvard, Boston, MA 02115, USA. [8] Instituto de Genética Médica y Molecular (INGEMM), Hospital Universitario La Paz-IdiPaz-UAM, 28046 Madrid, Spain. [9] Bone and Joint Research Unit, The Institution of Health Research (IIS)-Fundación Jiménez Díaz, UAM, 28040 Madrid, Spain. [10] Departamento de Anatomía, Histología y Neurociencia, Facultad de Medicina, Universidad Autónoma de Madrid, 28029 Madrid, Spain. [11] Servicio de Genética, Hospital Ramón y Cajal, 28034 Madrid, Spain. [12] Department of Clinical and Molecular Genetics and Rare Diseases Unit, Hospital Vall d'Hebron, 08035 Barcelona, Spain. [13] Centro Andaluz de Biología del Desarrollo (CSIC-UPO-JA), Universidad Pablo de Olavide, 41013 Sevilla, Spain. [14] Department of Molecular and Human Genetics, Jan and Dan Duncan Neurological Research Institute, Baylor College of Medicine, 1250 Moursund Street, Houston, TX 77030, USA. [15] Division of Clinical and Metabolic Genetics, Department of Paediatrics, The Hospital for Sick Children, University of Toronto, 555 University Avenue, Toronto, ON M5G 1X8, Canada. [16] Division of Neurology, Department of Paediatrics, The Hospital for Sick Children, University of Toronto, Toronto, ON M5G 1X8, Canada. [17]Present address: School of Pharmacy and Biomedical Sciences and Curtin Health Innovation Research Institute (CHIRI), Curtin University, Perth, WA 6102, Australia. These authors contributed equally: María Cristina Estañ, Elisa Fernández-Núñez. Correspondence and requests for materials should be addressed to G.Y. (email: grace.yoon@utoronto.ca) or to V.L.R.-P. (email: vlruiz@iib.uam.es)

Fragile X related 1 (FXR1), fragile X related 2 (FXR2), and fragile X mental retardation 1 (FMR1), comprise a family of homologous genes involved in posttranscriptional mRNA regulation. The protein products of these genes (FXR1P, FXR2P, and FMR1P, respectively) contain KH and RGG RNA-binding domains and have been shown to be associated with polysomal messenger ribonucleoparticles[1–4].

FXR1 is expressed in a wide variety of tissues and is subject to alternative splicing. Accordingly, seven different FXR1 mRNA isoforms (iso a-g) have been identified, some of which are tissue-specific[5,6]. Whereas most tissues express FXR1P variants ranging from 70 to 80 kDa, cardiac and skeletal muscle only generate FXR1P isoforms of 82 and 84 kDa (iso-e/f; P82,84), which contain an additional 81-nucleotide exon (exon-15) exclusive to these variants[5–8]. Testis is considered an exception since most FXR1P isoforms, including those carrying exon-15 were reported in this organ[9]. The incorporation of exon-15 into the P82,84 variants leads to the in frame insertion of 27 extra amino acids encompassing an arginine-rich motif that is also present in FXR2P, and is similar to the nucleolar localization signal of the HIV1-REV protein[10]. Both the distinctive splicing of exon-15 in muscle tissues and its corresponding amino acid sequence are conserved throughout vertebrate evolution[11,12]. However, expression of FXR1P proteins is not uniform among all cells of the myogenic lineage. During muscle differentiation, myoblasts synthesize all FXR1P isoforms, while myotubes only express P82,84 variants, although at higher proportion than their precursor cells[5,7,8].

Constitutive inactivation of Fxr1 in mice was reported to cause neonatal lethality most probably due to cardiac and/or respiratory failure secondary to loss of architectural structure of myofibers[13]. In the same manner, knockdown of Fxr1 resulted in muscle-specific defects in Xenopus and abnormal cardiac and striated muscle development in zebrafish[12,14]. Muscle cells from patients with facioscapulohumeral muscular dystrophy were also described with decreased P82,84 levels[7]. However, no human disease has previously been demonstrated to be associated with FXR1 mutations. Herein, we report that recessive mutations in exon-15 of FXR1 cause musculoskeletal defects of variable severity in humans and mice.

## Results

### Clinical evaluation.
Patients from two unrelated families were included in this study. The proband of family 1 was a 2.5-month-old Egyptian male, the second child of healthy first cousin parents. During pregnancy, reduced fetal movement was noted and serial ultrasound revealed oligohydramnios. This patient presented at birth with severe hypotonia, shallow breathing, episodes of tachycardia and fractures of the humeri and femora. On examination, he also had areflexia, tongue fasciculations, hypoplastic genitalia and cryptorchidism. Neurologically, the motor power was grade 1 (complete absence of movement). Brain CT and echocardiography were normal. The baby died at the age of five months. Medical termination of a subsequent pregnancy was conducted due to lack of fetal movement suggestive of the same condition. The affected status of this fetus (V-4) was subsequently clinically confirmed. Family history also included a similarly affected female sibling who died at the age of 70 days (Fig. 1a–c). MLPA of SMN1/2 in the proband excluded spinal muscular atrophy.

Patients from family 2 were three siblings with proximal muscle weakness, aged 28, 26, and 24 years, born to non-consanguineous parents. All three initially presented with neonatal hypotonia and delayed gross motor milestones (Fig. 1d). They also had moderate (II-3) or mild (II-4,-5) obstructive sleep apnea. The eldest sibling was additionally noted to have

cryptorchidism, short stature, obesity and mild scoliosis (13 degrees). All three siblings had normal cardiac and cognitive function. Mild progression of their muscle weakness occurred over the past five years, but all are independent with respect to activities of daily living, including ambulation. In all siblings electromyogram revealed low amplitudes and nerve conduction studies were normal, suggestive of a myopathic process. Detailed clinical information of families 1 and 2 is available as Supplemental Data.

### Muscular pathology of patients from family 2.
Histopathological analysis of muscle biopsy from II-4 revealed fatty infiltration and muscle fibers with increased central nuclei and striking type I fiber predominance (Fig. 1e–g). There was no fiber necrosis, regeneration or inflammatory infiltrates. Multiple small core lesions were detected within many muscle fibers which, by transmission electron microscopy (TEM), appeared as focal disruptions in the myofibrillar striation pattern, with Z-band streaming and absence of mitochondria. Cores were typically one to two sarcomeres in length (Fig. 1h). Muscle MRI for II-5 revealed extensive fatty atrophy of the anterior and posterior muscle groups.

### FXR1 exon-15 is mutated in patients with recessive congenital myopathy.
To identify the underlying genetic defect in the proband of family 1 we used homozygosity mapping and whole-exome sequencing (WES). Based on parental consanguinity, variants were filtered according to a recessive model of inheritance assuming a homozygous mutation by descent. This revealed a four-nucleotide deletion at the 3′-end of exon-15 of FXR1 (XM_005247813.3: c.1764_1767delACAG (delACAG)) which was prioritized as candidate due to the role of FXR1 in muscle development (Supplementary Table 1a,b). The four-nucleotide deletion causes a frameshift predicted to replace the last 90 amino acids of the muscle FXR1P isoforms (iso-e/f; P82,84) with 36 new residues (p.Arg588Serfs*37). Since the premature stop codon generated by this variant remains in the last exon of the gene, the corresponding mutant iso-e/f transcripts escape nonsense mediated mRNA decay (NMD) degradation[15] (Supplementary Fig. 1a). The delACAG variant was absent from population databases (gnomAD, ExaC, EVS, and 1000G) and was confirmed to be homozygous in the proband and in the affected fetus (V-4) by Sanger sequencing, while both parents were heterozygous (Fig. 1a). No DNA was available from the deceased sibling V-2. Consistent with parental consanguinity, the FXR1 variant was embedded within large chromosomal segments of homozygosity in the proband (54.6 Mb) and V-4 (23.7 Mb) (Supplementary Fig. 2a).

In family 2 we applied WES to four members of the family (the three affected siblings and the mother), which revealed a homozygous single nucleotide deletion also in exon-15 of FXR1 (XM_005247813.3: c.1707delA (delA); p.Lys569Asnfs*57) in all three affected siblings. The homozygous variant was validated by Sanger sequencing in II-3, II-4, and II-5 and the asymptomatic mother was confirmed to be a heterozygous carrier of the mutation (Fig. 1d). Even though the parents were apparently unrelated, the three affected siblings resulted to be homozygous for a 5.98 Mb region comprising FXR1 by SNP-array hybridization. Deletion of this region in the paternal allele was excluded following assessment of the SNP-array log-R ratio plots and by aCGH (Supplementary Fig. 2b, c).The c.1707delA deletion causes the same shift in reading frame as the c.1764_1767delACAG variant. However, c.1707delA is located at the 5′-end of exon-15, and thus the predicted P82,84 isoforms lack most exon-15-specific amino acids, (Supplementary Fig. 1a, c, d). Variants of

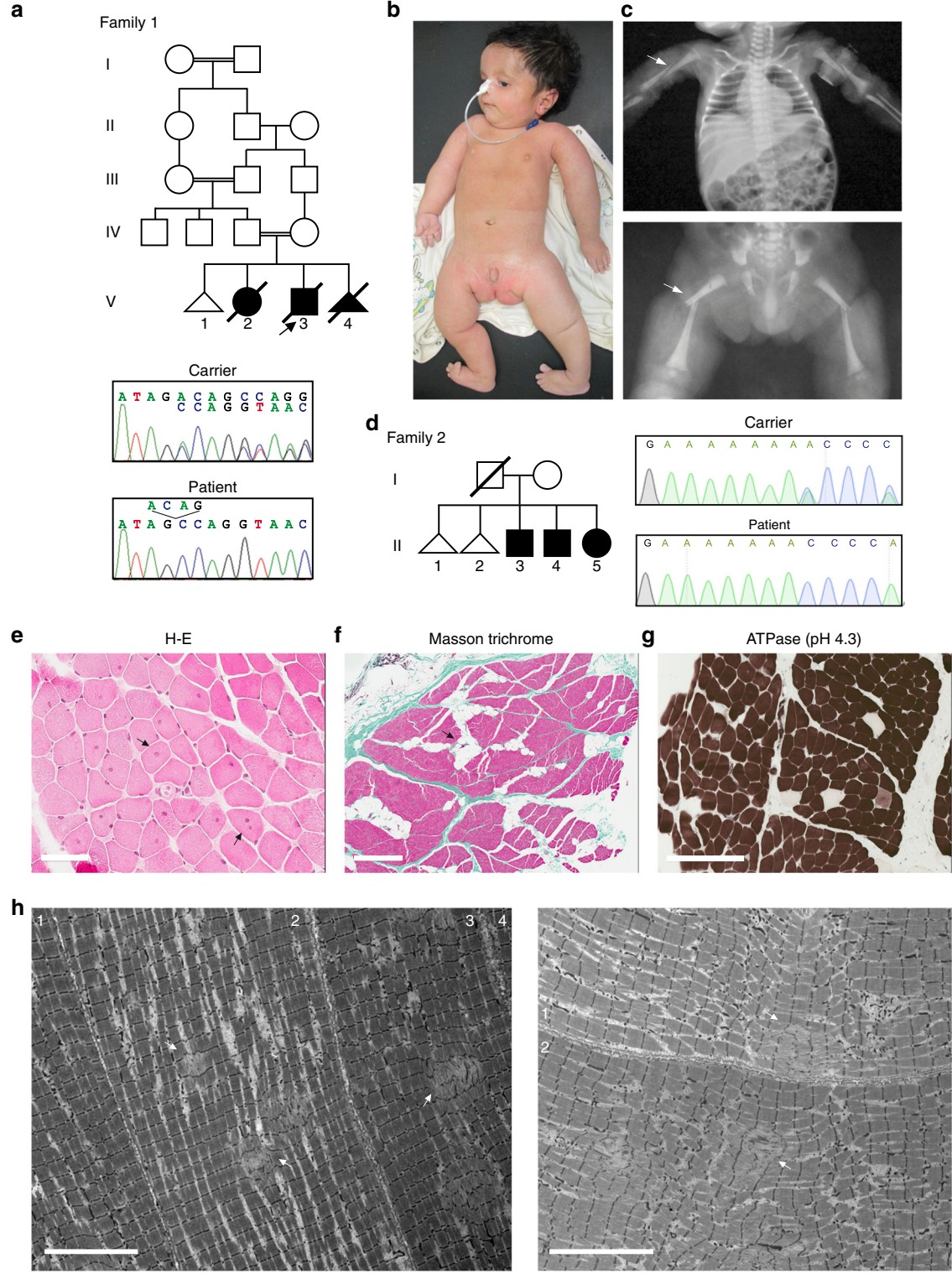

**DelACAG P82,84 variants localize to cytoplasmic granules**. To study the effect of the delACAG mutation on FXR1P proteins, we isolated primary myoblasts from V-4 and checked FXR1P expression in these cells before and after differentiation into myotubes. As predicted, western blot (WB) analysis demonstrated that V-4 cells synthetize P82,84 proteins of lower molecular weight. However, unlike control cultures, these variants were predominantly detected in the non-soluble fraction of both myoblast and myotube lysates. On the contrary, expression of P70-80 isoforms was normal in cell extracts from V-4 myoblasts (Fig. 2a and Supplementary Fig. 3). Consistent with this, the P82,84 isoforms from V-4 myotubes were localized in ring-shaped cytoplasmic granules by anti-FXR1P immuno-fluorescence, while in the corresponding control cells, the wild

genes that are known to cause multi-minicore myopathy and could act as potential genetic modifiers, identified in both families with allele frequencies less than 0.1, are indicated in Supplementary Table 2.

**Fig. 1** Clinical features and mutations. **a** Pedigree of Family 1 showing the proband (arrow), similarly affected older female sibling and affected fetus. DNA-sequencing chromatograms demonstrate a homozygous four nucleotide deletion at the 3′-end of *FXR1* exon-15 in the proband, which is in the heterozygous state in his mother (carrier). Carrier chromatogram shows normal (top) and mutant (underneath) nucleotide sequences. **b** Proband of family 1 at the age of 2.5 months with inserted Ryle tube for feeding, short neck, short hands and digits, lateral rotation of right upper and lower limbs and medial rotation of left upper and lower limbs due to severe hypotonia. The picture also shows bilateral low inserted thumbs with dorsi-flexion of both feet, no obvious demarcation of large joints with transverse crease on the left knee and ankles and genital hypoplasia. **c** X-rays (AP view) showing bilateral mid fractures of humeri (upper panel, arrow) and femora (lower panel, arrow). **d** Pedigree of family 2 and DNA-sequencing chromatograms corresponding to the 5′-end of *FXR1* exon-15 of the mother (carrier) and one of the affected siblings showing a single adenine deletion from a run of eight-adenines in the homozygous state in the patient. The father was not available for carrier testing. Only nucleotide sequence of the wt allele is written on the chromatogram of the mother. **e–h** Representative photomicrographs of H-E staining. Scale bar 60 μm (**e**), Masson trichrome staining. Scale bar 500 μm (**f**), ATPase pH4.3 histochemistry. Scale bar 200 μm (**g**), and TEM. Scale bar 10 μm (**h**) from a right triceps biopsy of individual II-4 of family 2. Arrows indicate internalized nuclei, fatty infiltration, and areas of Z-streaming and minicores in **e**, **f** and **h**, respectively. Different fibers in TEM images are numbered

type (wt) P82,84 proteins were uniformly distributed throughout the entire cytoplasm. Anti-FXR1P staining in myoblasts, which generate less amount of P82,84, revealed similar cytosolic distribution of FXR1P isoforms in V-4 and control cells, except for a small number of FXR1P granules in some V-4 myoblasts (Fig. 2b). No differences in levels of FXR1P isoforms were observed between patient and control primary fibroblasts (Supplementary Fig. 3).

**Myf5-dependent inactivation of Fxr1 is lethal in mice**. Constitutive ablation of all FXR1P isoforms in mice due to *Fxr1* exon-1 deletion was previously shown to be neonatal lethal[13]. Although not definitive, this phenotype was attributed to skeletal and/or cardiac muscular defects. To investigate this further we generated tissue-specific knockout mice in which the entire repertoire of FXR1P isoforms was primarily inactivated in skeletal muscle. This was accomplished by mating *Fxr1^cko/cko* floxed mice carrying the first exon of *Fxr1* flanked by *LoxP* sites to *Myf5-Cre* transgenic mice (B195AP-Cre)[16] that were made heterozygous for a constitutively inactivated *Fxr1* allele lacking exon-1 (*Myf5-Cre^+/−*; *Fxr1^+/ko*). *Myf5* codes for the earliest myogenic transcription factor that directs progenitor cells to differentiate into the skeletal muscle lineage and is involved in myoblast specification[17,18]. Evaluation of litters from these crosses at P0-1 identified 19% neonatal lethality with the large majority of non-surviving newborn mice having the *Fxr1^ko/cko*;*Myf5-Cre^+/−* genotype (87.5%). In contrast, only 1.51% of all living pups showed this allelic combination (Fig. 3a and Supplementary Fig. 4a, b). Specific ablation of FXR1P in skeletal muscle of *Fxr1^ko/cko*;*Myf5-Cre^+/−* mice was confirmed by WB using extracts from several tissues of E18.5 mice (Supplementary Fig. 4c, d). Hence, loss of the complete collection of FXR1P isoforms in the skeletal myogenic lineage alone is sufficient to cause neonatal lethality.

**Mice with exon-15 mutations display myopathic phenotypes**. We replicated the delACAG variant in mice by CRISPR-Cas9 genomic editing. Human and mouse *FXR1* are 95% identical at the DNA level and the corresponding delACAG deletion in the mouse (NM_001113188.1:c.1764_1767delACAG; p.(Arg588-Serfs*37) also result in the substitution of the last 90 amino acids of the murine iso-e/f by the same 36 extra amino acids of the mutant protein of family 1. In parallel, another mouse line harboring a different exon-15 mutation consisting of one nucleotide duplication was generated during genomic editing (NM_001113188.1:c.1766dupA). The new frameshift resulting from this allele runs into a premature termination codon 10 triplets downstream within exon-16 (p.Pro590Alafs*10) and is predicted to undergo NMD (Supplementary Fig. 1).

Homozygous mutants of the two CRISPR lines were born at Mendelian ratio and survived to adulthood, but developed a

muscular phenotype which was more severe in delACAG mice. On gross observation, delACAG homozygotes (delACAG) were leaner than their littermates and exhibited lower body weight than their corresponding sex-matched and age-matched wt controls (mean body weight reduction of delACAG females at age 6, 9, and 12 weeks: −15.56%, −20.94%, and −16.43%, respectively). No phenotype was observed in delACAG heterozygous mice. Homozygous dupA mutants (dupA) were macroscopically indistinguishable from their littermates, but also had decreased body weight compared to controls (mean body weight reduction of dupA females aged 6, 9, and 12 weeks: −2.53%, −7.58%, and −6.97%, respectively) (Fig. 3b, c). Following dissection of hindlimbs and gastrocnemius, it became apparent that delACAG mutants had reduced muscle mass (Fig. 3d). The muscle phenotypes of delACAG and dupA mice were confirmed by MRI of hindlimbs of 14-week-old female mice (Fig. 3e and Supplementary Fig. 4e). Additionally, we evaluated muscle strength in both exon-15 mutants by comparing the weightlifting capacity of mice and by the hanging wire test, while sustained rotarod exercise was used to measure fatigue resistance. In all tests, delACAG mice scored significantly lower than their wt littermates, while dupA mice performed similarly or slightly less well than controls (Fig. 3f, g and Supplementary Fig. 4f).

**Bone mineral density is decreased in delACAG mice**. Bone mineral density (BMD) was assessed in both CRISPR mouse models using DXA. BMD calculated both at the femur and at the lumbar vertebrae resulted to be significantly lower in delACAG mutants compared to their wt littermates. Milder BMD differences between dupA and control mice were also observed, but were not statistically significant (Fig. 3h). Radiographs of dissected mouse limbs additionally demonstrated decreased BMD in delACAG mice (Fig. 3i and Supplementary Fig. 4g).

**Multicore myopathy in mice carrying exon-15 mutations**. We performed histological characterization of murine delACAG and dupA muscles using transverse frozen sections of vastus lateralis/gastrocnemius/soleus. H–E staining revealed reduced fiber size, increased cell density (cells/area) and increased central nuclei in mutant mice, with the severity of these defects being greater in delACAG homozygotes. The average length of minimal Feret's diameter (MFD) determined in the vastus lateralis was reduced by 41.56% in delACAG mutants and 23.11% in dupA mice, and the percentage of fibers with central nuclei in the same muscle was 31.87% and 13.41% in delACAG and dupA mutants, respectively. NADH-TR histochemistry identified areas with no enzymatic activity (cores) in multiple delACAG fibers, while cores were very sporadically detected in dupA mice. Lastly, ATPase histoenzymatic staining (pH 4.3) revealed predominance of type I fibers in both mutants, especially in delACAG mice.

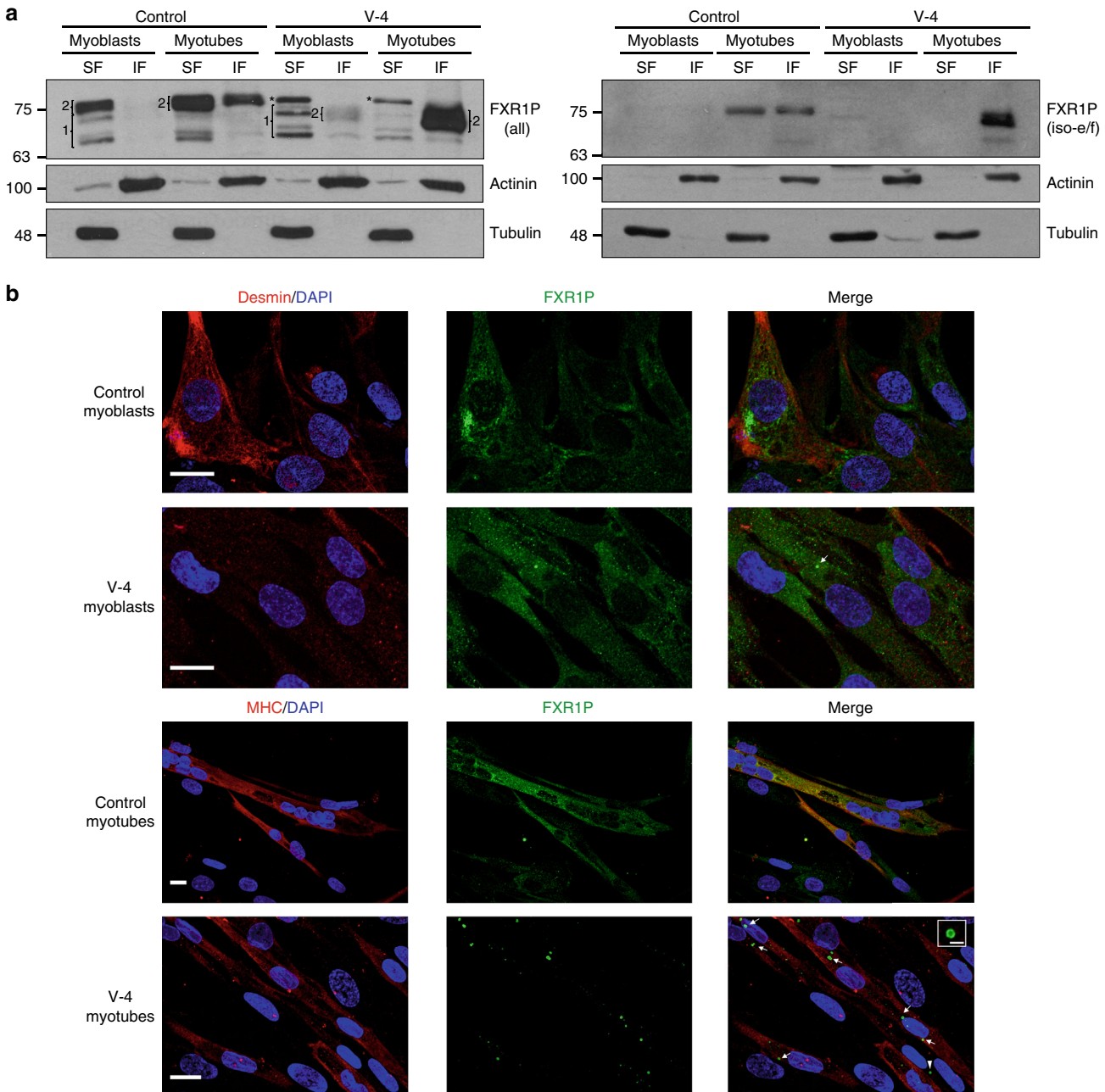

**Fig. 2** Expression and localization of FXR1P isoforms in family 1. **a** Representative anti-FXR1P immunoblots of soluble (SF) and insoluble (IF) fractions of protein extracts from control and V-4 myoblasts and myotubes, $n = 5$. An antibody against all FXR1P isoforms (Proteintech) and another one specific for iso-e/f (#27-15) were used in left and right panels, respectively. Numbered brackets on left panel designate P70-80 FXR1P isoforms (1), and P82,84 isoforms (2). Asterisks indicate a non-specific band. α-actinin and α-tubulin were used as loading controls. **b** Anti-FXR1P immunofluorescence showing subcellular distribution of FXR1P in control and V-4 myoblasts and myotubes, $n = 5$. Scale bar 15 μm. Arrows in V-4 myoblasts and myotubes point to ring-shaped granular accumulations of mutant P82,84 isoforms. The granule magnified in V-4 myotubes is indicated with an arrowhead (scale bar 2 μm). Desmin and MHC were used as myogenic markers with MHC being an indicator of myotube differentiation. Nuclei are stained with DAPI

Relative frequency of type I fibers in the soleus was 43.3% in wt, 90.4% in delACAG and 53.3% in dupA mutants (Fig. 4a–e).

Muscle fibers from the gastrocnemius of exon-15 mutants and control mice were analyzed by TEM. Consistent with NADH-TR staining, the predominant pathological feature of delACAG fibers consisted of intracellular regions of variable size characterized by disintegration of Z-band and sarcomere structure (cores). Ultrastructurally, the appearance of delACAG fibers ranged from mildly affected fibers containing only several regions of Z-streaming to highly disorganized cells with a large core

spread over nearly the entire fiber section. Misalignment, irregular shortening, broadening and zig-zagging of Z-lines were observed in zones with preservation of sarcomeres. In addition, abnormal accumulations of mitochondria and central nuclei were detected. TEM analysis of dupA fibers showed abnormalities similar to those observed in delACAG mice with respect to the appearance of Z-lines and there were occasional small areas of Z-streaming. Cores were not detected suggesting that they are infrequent in these mice (Fig. 4f, g and Supplementary Fig. 5a, b).

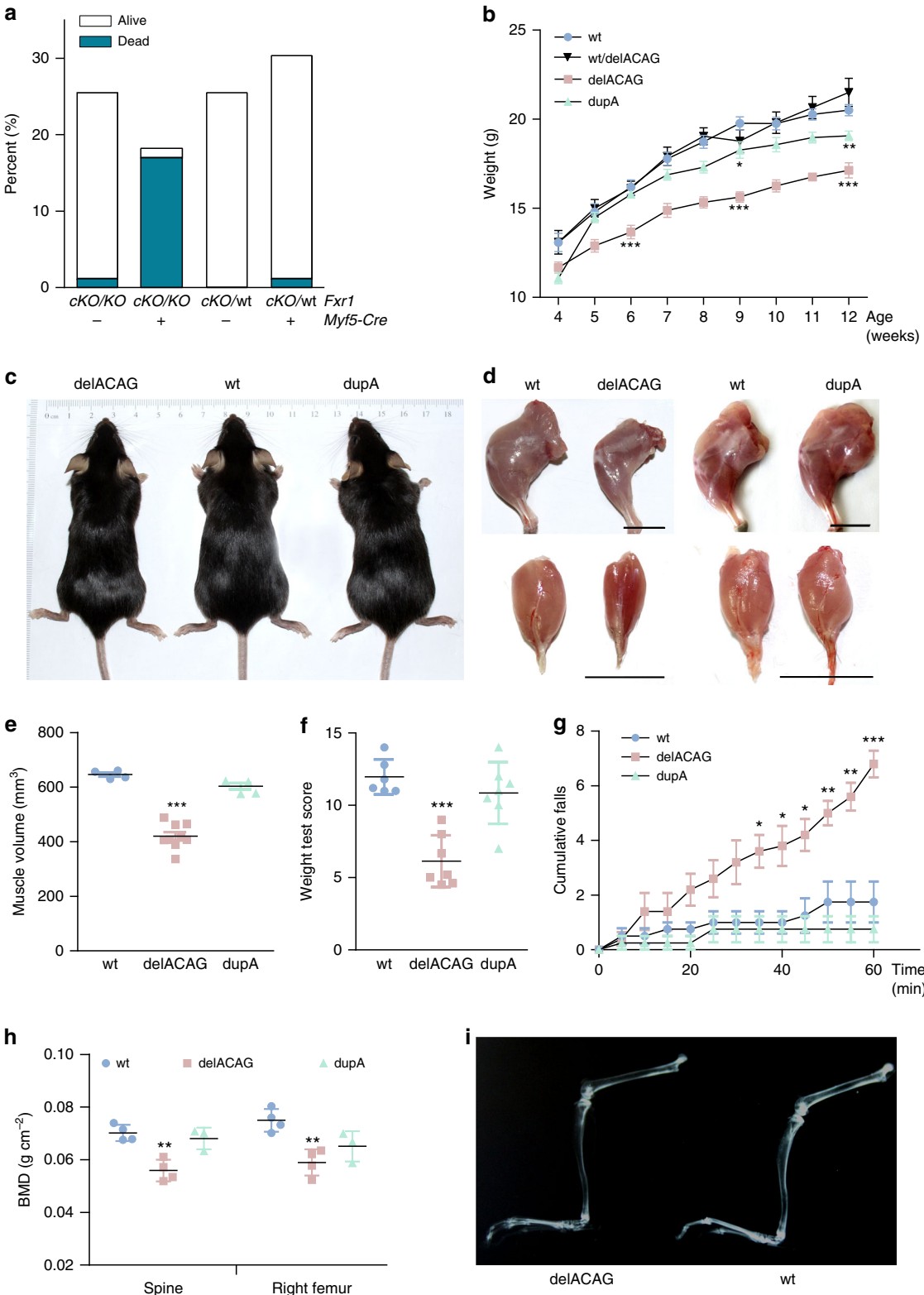

**Expression and localization of P82,84 in exon-15 mutants**. To understand the phenotypic differences between delACAG and dupA mice, we studied *Fxr1* expression both at the mRNA and protein levels in skeletal muscle of these mutants. In keeping with the dupA transcript being subjected to NMD, qRT-PCR analysis showed that the levels of *Fxr1* mRNA in dupA mice were reduced to 30.9% of control levels. The same experiment in delACAG mice revealed a milder reduction of the *Fxr1* transcript to 74.7% of control levels (Fig. 5a and Supplementary Fig. 6). Since the delACAG mutation should not trigger NMD, this data could indicate the disruption of a positive feedback mechanism. Muscle expression of FXR1P proteins was analyzed by immunoblotting. This demonstrated a marked decrease in the levels of FXR1P isoforms in dupA mice, with just a faint band corresponding to the dupA P82,84 molecular weight detectable in the soluble fraction of muscle extracts. In delACAG mice the quantity of

**Fig. 3** Phenotype of mice with different *Fxr1* mutations. **a** Percentage (%) of neonatal lethality at P0-1 in offspring ($n = 82$) from crosses between *Myf5-Cre*[+/−];*Fxr1*[+/ko] males and *Fxr1*[cko/cko] females. Genotypes for *Fxr1* and *Myf5-Cre* are indicated in the *X*-axis. Presence or absence of *Myf5-Cre* is indicated with + and – symbols. **b** Growth chart of homozygous delACAG and dupA mice compared to wild type controls (wt) and wt/delACAG heterozygotes. Female mice were weighed weekly after weaning from week 4 to 12. $n = 16$(wt), 13(wt/delACAG heterozygotes), 12(delACAG), 9(dupA). Data are mean ± s.e.m. followed by two-tailed unpaired Student's *t*-test. **c** Representative image of three months old delACAG, wt and dupA female mice. **d** Representative images of left hindlimb (top panel) and gastrocnemius (bottom panel) from delACAG and dupA mutants compared to their respective wt littermates (3-month old female mice). Scale bar 1 cm. **e** Muscle volume corresponding to a segment of hindlimbs determined by MRI (14 weeks-old female mice), $n = 4$(wt), 9(delACAG), 4(dupA). Data are mean ± s.d. followed by one-way ANOVA (***) with Tukey post-hoc test. **f** Score values for weight test (14 weeks-old female mice, $n = 6$(wt), 7(delACAG), 7(dupA)). Data are mean ± s.d. followed by one-way ANOVA (***) with Tukey post-hoc test. **g** Cumulative number of falls during rotarod endurance test (14 weeks-old female mice), $n = 4$(wt), 5(delACAG), 4(dupA). Data are mean ± s.e.m. followed by one-way ANOVA (**) with Tukey post-hoc test. **h** BMD calculated by DXA at the spine and the right femur of 4 months old female mice, $n = 4$ (wt), 4(delACAG), 3(dupA). Data are mean ± s.d. followed by one-way ANOVA (**) with Tukey post-hoc test. **i** Representative X-Ray image corresponding to dissected left hindlimbs from delACAG and wt littermates (4 months-old female mice), $n = 4$(wt), 1(wt/delACAG), 8(delACAG), 4 (dupA)

P82,84 appeared not to be substantially diminished but, as in V-4 myotubes, these variants were principally localized in the non-soluble part of muscular lysates (Fig. 5b). The same results were observed on WB of primary myogenic cultures from these mice (Supplementary Fig. 7a). Subsequently, we used confocal microscopy in isolated fibers from the extensor digitorum longus (EDL) to ascertain the subcellular localization of normal and mutant P82,84 isoforms. FXR1P immunofluorescence identified the wt P82,84 in a striped pattern perpendicular to the main axis of the fiber. This staining was strongly reduced in dupA fibers, while in delACAG fibers, similar to V-4 myotubes, the FXR1P proteins were detected in cytoplasmic granules of variable dimensions (Fig. 5c). Immunofluorescence analysis of mouse primary cultures showed dispersed cytoplasmic distribution of FXR1P isoforms in all genotypes at the myoblast stage with presence of ring-shaped V-4-like granules exclusively in delACAG cultures. Following differentiation, FXR1P expression was essentially absent in dupA myotubes and was restricted to ring-shaped aggregates in delACAG differentiated cells (Supplementary Fig. 7b, c).

Subcellular distribution of P82,84 in mouse striated muscle has previously been described as a dot-like pattern reminiscent of costameres associated with Z-bands, but robust co-localization analysis had not been performed[5,12,13]. To refine the localization of normal P82,84 isoforms and to compare it with that of the mutant proteins we conducted double-immunostaining experiments using recognized sarcomere markers. Co-immunofluorescence of FXR1P with α-actinin or phalloidin to visualize Z and M lines showed the wt FXR1P isoforms in a position immediately adjacent to Z-lines. Consistently, double-immunostaining of FXR1P with RYR and COX-IV, as markers for triads (T-tubule + sarcoplasmic reticulum (SR)) and mitochondria, respectively, identified the P82,84 wt proteins between the two triads that flank each Z-line, and partially overlapping with mitochondria. Subcellular localization of the remaining P82,84 isoforms in dupA fibers was comparable to the wt proteins, while the FXR1P signal was almost absent from Z-lines in delACAG fibers (Fig. 6). Anti-COX-IV immunofluorescence demonstrated both mitochondrial accumulations and areas deprived of mitochondria (cores) in delACAG fibers, with a proportion of P82,84 granules overlapping or placed near mitochondrial groupings. Fewer accumulations of mitochondria were detected in dupA fibers (Fig. 6b and Supplementary Fig. 8a).

Next, we examined ultrathin sections of murine gastrocnemius by TEM to search for delACAG granules. We identified a series of non-membranous globular/ring-shaped cytoplasmic structures of heterogeneous size in delACAG mutants, but not in wt or dupA mice (Fig. 7a). TEM analysis of V-4 and control differentiated myotubes at the EM level also revealed similar type of granular structures in patient cells (Fig. 7b). Ultrastructurally, delACAG granules were composed of a highly electrodense

external layer encircling a different kind of material. Both, TEM and confocal FXR1P-positive granular structures had similar diameter length distribution (mean MFD: 0.51 μm (TEM) and 0.59 μm (confocal)) and were often found next to mitochondria (Fig. 7a and Supplementary Fig. 8b). Immunogold labeling of FXR1P on gastrocnemius sections conclusively identified the mutant protein in the outer electrodense layer of delACAG granules (Fig. 7c, d). Immunofluorescence and TEM analysis of fibers from wt/delACAG heterozygotes detected similar granular structures, although in a much smaller number than in delACAG homozygotes (Fig. 5c and Supplementary Fig. 8c). In contrast, as in dupA mice, no FXR1P granules were observed by TEM in the muscle biopsy of family 2.

**DelACAG granules contain polyA-mRNA but differ from SGs.** Since delACAG FXR1P isoforms maintain all the RNA binding motifs of the wt protein, we investigated whether they could interact with polyA-mRNA. We conducted fluorescence in situ hybridization (FISH) in isolated EDL mouse fibers and V-4 myotubes using a Cy3-labeled oligo-dT probe and observed that a proportion of delACAG granules, commonly the bulkiest, were positive for oligo-dT hybridization. In contrast, no signal was obtained with a similarly labeled control probe consisting of a scrambled nucleotide sequence with no specific target[19]. Thus, delACAG granules incorporate mRNA (Fig. 8 and Supplementary Fig. 9).

Consequently, we analyzed if delACAG accumulations could function as stress granules (SGs) given that FXR1P is a component of these structures[20]. SGs are RNA-protein aggregates triggered by cellular stress that stall mRNA translation[21,22]. To test this hypothesis, we co-immunostained V-4 and control myoblasts with FXR1P and the SG markers TIAR and FXR2P[20,23]. Myoblasts were cultured in differentiation media for one day to promote iso-e/f expression and subsequently maintained untreated or stressed with arsenite (ARS). Complete staining overlap between FXR1P and TIAR or FXR2P was observed in ARS-induced amorphous granules corresponding to SGs, both in control and patient myoblasts. However, there was only faint or no co-localization between FXR1P and SG markers in characteristic ring-shaped delACAG granules. Co-immunostaining of delACAG particles with processing bodies (P-bodies), which are a different type of RNA-protein aggregates involved in mRNA degradation[21,22] was also examined. This was achieved by FXR1P and RCK double immunolabelling, with RCK being a marker for both P-bodies and SGs[24]. Again, no-co-localization was observed between FXR1P and RCK at ring-shaped delACAG granules indicating that they are not P-bodies (Fig. 9a, b).

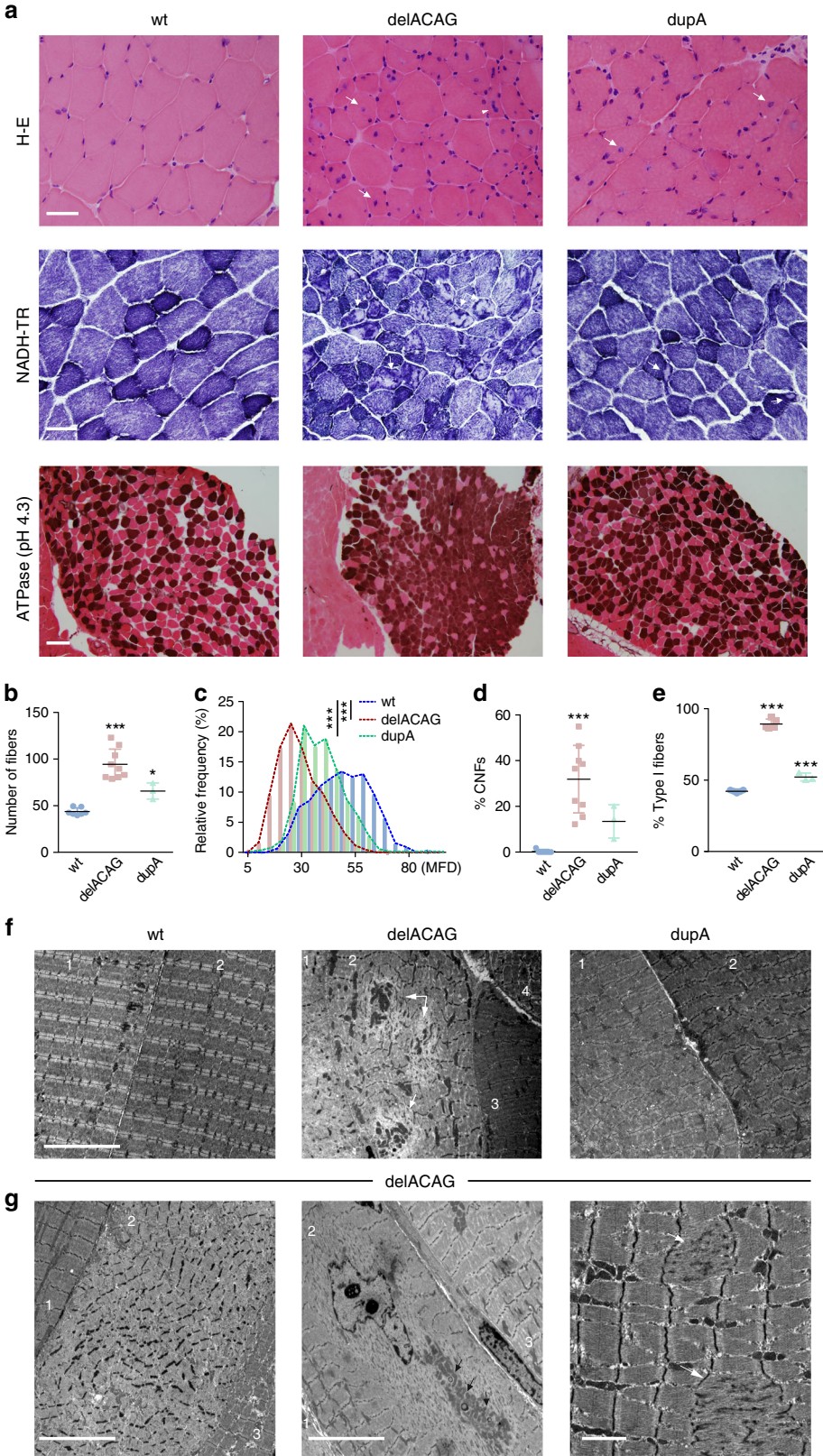

**Mutant iso-e variants assemble into delACAG granules in HeLa.** We checked if delACAG granules can assemble outside of a myogenic environment, by transfecting HeLa with N-terminally EGFP tagged human iso-e constructs carrying either wt sequence or the delACAG, delA or dupA mutations. Expression of all fusion proteins was validated by WB in HEK293T (Supplementary Fig. 10a, b). Moreover, to evaluate the involvement of iso-e in SGs, we treated the transfected HeLa with or without ARS and analyzed the expression of EGFP and SG markers, including TIAR, FXR2P and FMR1P[20]. Robust co-localization between wt iso-e and SG markers was observed in ARS-treated cells, thus proving that this isoform is recruited into

**Fig. 4** Multicore myopathy in mice carrying exon-15 mutations. **a** Top panel: H-E staining of vastus lateralis cryosections. Arrows designate central nulei and the arrowhead in delACAG indicates three centralized nuclei. Middle panel: NADH-TR histochemistry of vastus cryosections showing multiple areas with no enzymatic activity (cores) in delACAG mice and very few in dupA mice. Arrows depict examples of cores. For both panels: n = 7(wt), 9(delACAG), 3(dupA) and scale bars 50 µm. Lower panel: ATPase (pH 4.3) histochemistry in whole soleus cryosections revealing increased type 1 fibers in both mutants, n = 5(wt), 6(delACAG), 3(dupA). Scale bar 0.5 mm. **b** Quantification of fiber density (number of fibers/microscope field of view). Graph represents mean ± s.d. **c** Minimal Feret's diameter (MFD, µm) showing relative frequency (%) of different fiber sizes with respect to the total number of fibers analyzed. Differences in mean MFD between wt and homozygous mutants were statistically significant (wt: 49.8 µm ± 12.9 s.d., delACAG: 29.1 µm ± 9.9 s.d., dupA: 38.3 µm ± 9.7 s.d.). **d** Percentage of fibers with central nuclei (% CNFs) in wt, delACAG and dupA mice. Graph corresponds to mean ± s.d. For **b–d** H-E cryosections of vastus lateralis were used and a total number of n = 455(wt), 1726(delACAG), 465(dupA) fibers were analyzed from 7(wt), 9 (delACAG) and 3(dupA) mice. **e** Percentage (%) of type I fibers calculated in soleus sections expressed as mean ± s.d. n = 5(wt), 5(delACAG), 3(dupA) mice. One-way ANOVA (***) with Tukey post-hoc test was used in **b** and **e** and Kruskal-Wallis (***) with Dunn's multiple comparisons test in **c** and **d**. **f** Representative TEM images of gastrocnemius from wt, delACAG and dupA mice, n = 3(wt), 4(delACAG), 2(dupA) mice. Scale bar 10 µm. Numbers identify different fibers. Note the variable disorganization of delACAG fibers. Arrows indicate cores. **g** TEM images of gastrocnemius showing different defects in delACAG fibers including: large cores (left and middle image. Scale bars 10 µm), central nuclei and abnormal mitochondrial accumulations (middle image), and Z-line abnormalities including Z-streaming (arrows, right image. Scale bar 2 µm). Arrows in the central picture point to ring-shaped granules shown in Fig. 7a

SGs. In untreated cells, wt iso-e was found disseminated in the cytoplasm, although we noted a variable proportion of stressed cells, (owing to transfection or FXR1P overexpression) with iso-e in SGs. However, independent of ARS treatment, all three mutant iso-e proteins (delACAG, delA, and dupA) were located in cytoplasmic granules either as part of amorphous SGs or assembled into delACAG-like ring-shaped aggregates (Fig. 9c and Supplementary Fig. 10c). HeLa delACAG-like granules exhibited no overt co-localization with TIAR or FMR1P, and only a few had faint positive co-localization with FXR2P (Supplementary Fig. 11). As the extra C-terminal amino acids incorporated after the dupA mutation differ from those added by the delA/delA-CAG mutations (Supplementary Fig. 1d), the accumulation of mutant iso-e into delACAG-like granules in HeLa does not depend on a specific C-terminal amino acid sequence. Absence of granules in dupA mice and in the muscle biopsy of family 2 is likely due to low expression levels of the corresponding P82,84 as demonstrated in muscle tissue of dupA mice. No biological material was available from family 2 to test this hypothesis.

**Expression profiling of delACAG skeletal muscle.** To investigate the molecular processes disrupted in delACAG mice we conducted comparative RNA-Seq-based transcriptome analysis of delACAG and control muscles using RNA from gastrocnemius + soleus of P15 mice. Following data normalization 233 differentially expressed genes (DEGs) were selected of which 150 were upregulated and 83 downregulated in the mutant mice. Selected DEGs had fold changes ≥1.9 or ≤−1.9 (p-adjusted value ≤ 0.05). Gene ontology (GO) functional enrichment analysis of these transcripts yielded 25 GO annotations including p53 signaling, G1/S cell cycle transition, DNA damage response; apoptosis, muscle-development, muscle-differentiation, and muscle-contraction, protein kinase B signaling, action potential and membrane depolarization (Fig. 10a–c and Supplementary Table 3a). Moreover, within the nominated DEGs we recognized known proinflammatory genes. KEGG enrichment analysis also identified p53 signaling as the most statistically significant pathway (Supplementary Table 3b). RNAseq results of representative genes were verified in gastrocnemius from delACAG mice by qRT-PCR, using *Tbp* and *Gusb* as reference genes, or WB analysis. In these experiments age-matched dupA mice were included for comparison (Fig. 10d–f and Supplementary Fig. 12). Quantification of the expression of the *p53/p63/p73* gene family by qRT-PCR revealed a remarkable induction in the levels of *Trp73* mRNA and to a lesser degree of the *Trp63* transcript in delACAG mice (Fig. 10d and Supplementary Fig. 12a). In contrast, two TaqMan assays showed no significant changes in *Trp53* mRNA

levels. Similarly, we confirmed upregulation in delACAG mutants of the following transcripts: (i) the cell cycle regulator *Cdkn1a* (ii) the proinflammatory/proapoptotic related transcripts *Nfkb2*, *Eda2r*, *Bex1*, *Cxcl10*, *Wdr35*, *Tnfrsf22* and *Tnfrsf23*; (iii) the transcription factor *Ankrd1*; and (iv) the embryonic myosin *Myh3* (Fig. 10d and Supplementary Fig. 12a). Increased expression of the apoptosis mediator caspase-3 in delACAG muscular tissue was confirmed by WB, although the levels of its processed active form varied between mice. Of note, a link between WDR35 and caspase-3 has been reported[25]. Overexpression of CDKN1A and ANKRD1 was corroborated by WB (Fig. 10e and Supplementary Fig. 12b, c). Two transcripts involved in muscle contraction, *Sln* and *Nos1*, were verified to be markedly up-regulated and down-regulated, respectively in delACAG mutants. Increased SLN levels were further proved by WB in the soleus and gastrocnemius of delACAG mice. Similarly, the amount of triadin (TRDN), which like SLN, is involved in $Ca^{+2}$ homeostasis was higher in muscle extracts from delACAG mutants than in control mice (Fig. 10f and Supplementary Fig. 12b, c). DelACAG muscles were also confirmed to have increased expression of the stress-related myokine *Gdf15* and reduced mRNA levels of the inhibitory myokine of muscle growth myostatin (*Mstn*). Equivalent qRT-PCR analysis of dupA mutants revealed a pattern of gene expression changes similar to delACAG mice, although for most genes the extent of these changes was lower or not statistically significant. Finally, we quantified Terc RNA in both exon-15 mutants since it was reported to be stabilized by FXR1P[26] and was absent in our RNAseq assay because lacks poly A tail. However, no significant differences were detected in the muscle expression of *Terc* between control and mutant mice (Fig. 10d and Supplementary Fig. 12a).

## Discussion
Herein, we show that recessive mutations in exon-15 of *FXR1* are associated with a novel multi-minicore myopathy in humans and mice that varies in severity depending on the consequences of each mutation. This work also sheds light on the functional relevance of the evolutionary conserved P82,84 variants in comparison to the rest of FXR1P isoforms. Additionally, we demonstrate that certain *FXR1* mutations lead to the synthesis of abnormal P82,84 proteins which accumulate in a new type of granular structures inside the cell. DelACAG granules incorporate mRNA, but are distinct from SGs. Thus, they likely correspond to misfolding and subsequent aggregation of mutant FXR1P isoforms as described for other proteins in some neuro-degenerative diseases[27].

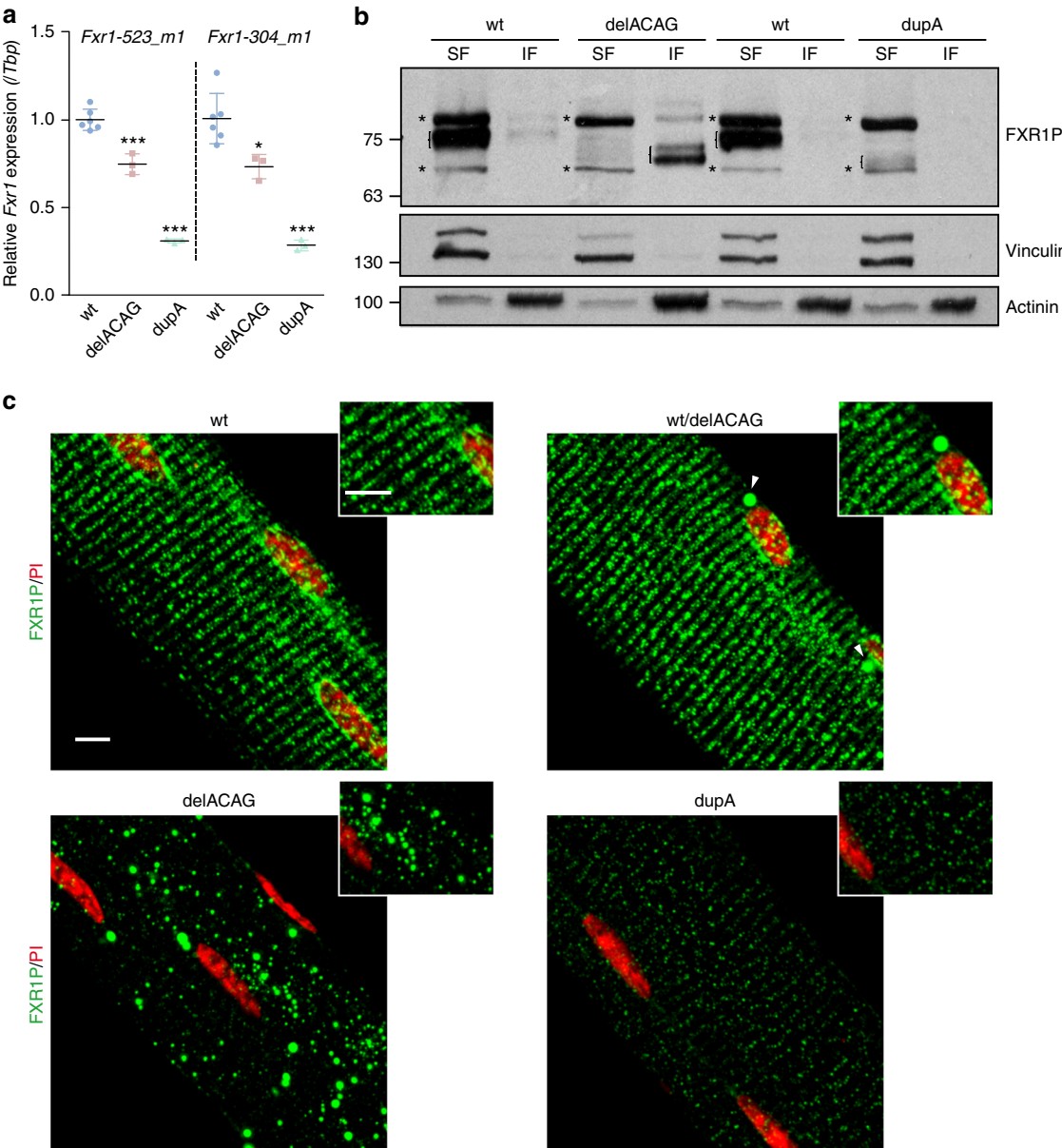

**Fig. 5** Expression and subcellular localization of P82,84 isoforms in exon-15 mice. **a** Relative *Fxr1* mRNA quantification in gastrocnemius of 2.5-months old mice by qRT-PCR using two different TaqMan probes: *Fxr1-523_m1* (exons 2-3; Mm00484523_m1) and *Fxr1-304_m1* (exons 14-15; Mm01286304_m1). Values are normalized to *Tbp* mRNA levels and represented as fold change of the mean value of wt mice. $n = 6$ (wt), 3(delACAG), 3(dupA). Data are mean ± s.d. followed by one-way ANOVA (***) with Tukey post-hoc test. **b** Representative FXR1P immunoblot of protein extracts from gastrocnemius (soluble (SF) and insoluble (IF) fractions) of delACAG and dupA mice and corresponding wt littermates. Vinculin and α-actinin were used as loading controls (2–3 months old mice, $n = 4$). Brackets denote isoforms e and f and non-specific bands are labeled with asterisks. **c** Maximum Z-project and magnification of confocal anti-FXR1P immunofluorescence images (green) in isolated EDL fibers. $n = 8$(wt), 3(wt/delACAG), 8(delACAG), 6(dupA). Scale bars 5 μm. FXR1P isoforms are localized in a striated pattern except in delACAG fibers where are found predominantly in granules. Heterozygous wt/delACAG fibers contain a small number of granules (arrowheads) and the fluorescent intensity of FXR1P is reduced in dupA mice. Nuclei are labeled with propidium iodide (PI, red)

Our data indicate that *Myf5-Cre* dependent depletion of all FXR1P isoforms is neonatal lethal, while mice with mutations exclusively disrupting P82,P84 are viable, even if the dupA/delACAG mutations are associated with marked reduction of P82,84 protein levels (dupA), or with abnormal P82,84 subcellular localization (delACAG). Therefore, we conclude that loss of P70-80 FXR1P isoforms, which are specifically expressed in early myogenic progenitors, accounts for the lethal phenotype of the *Myf5-Cre;Fxr1* conditional knockouts, while P82,84 are required for maintaining myofilament alignment and organization of Z-lines. It has been suggested that similar to the role of FMR1P in neurons, FXR1P could maintain specific mRNAs in a repressed state in muscle costamere structures until they are required to be translated for de novo protein synthesis[12]. Consistent with this hypothesis we observed that FXR1P-iso-e is recruited into SGs which repress translation. Hence, deregulated translation of specific mRNAs involved in Z-line organization could underlie the minicore phenotypes resulting from P82,84 mutations.

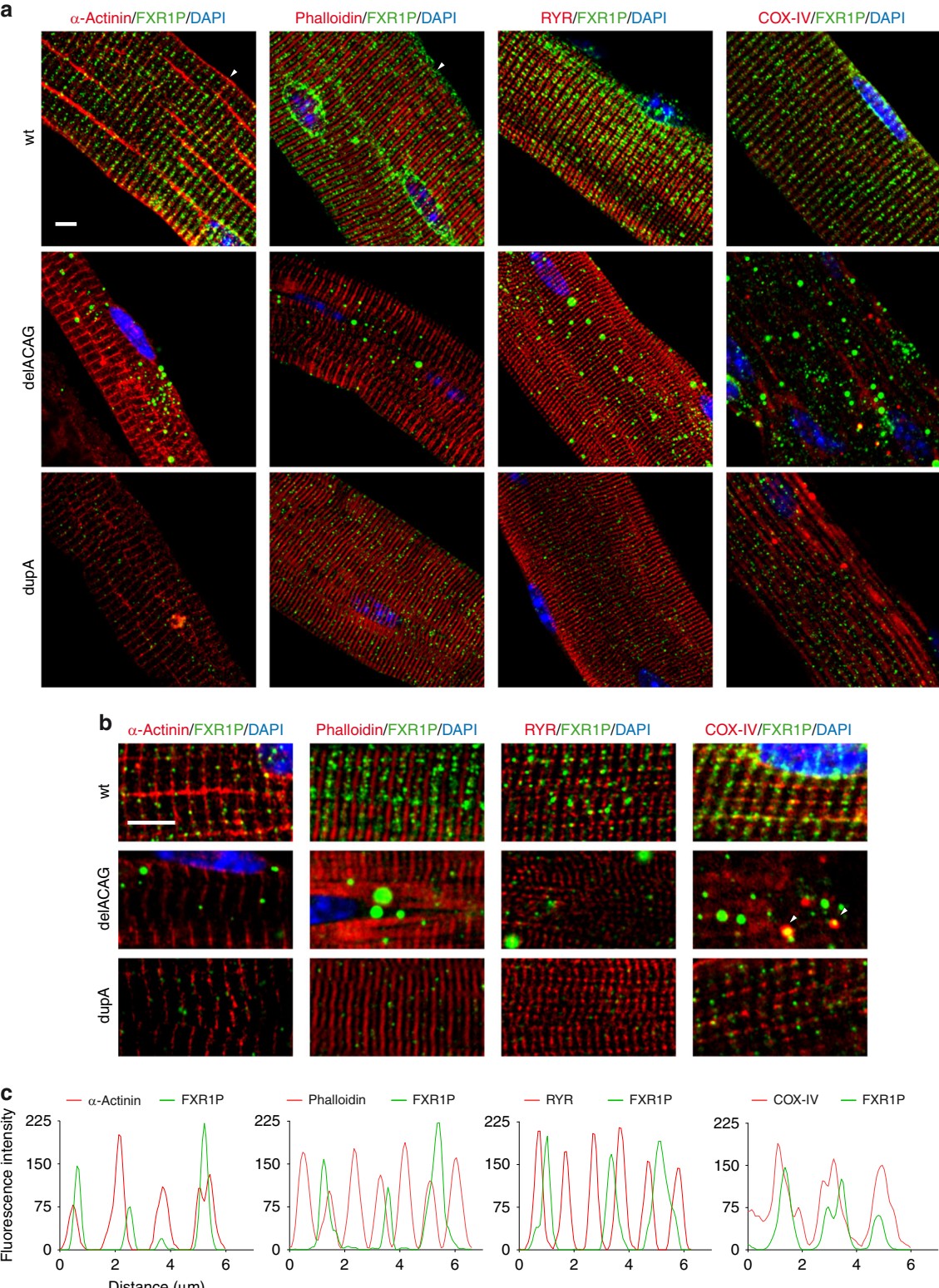

**Fig. 6** Co-localization analysis of P82,84 isoforms in EDL fibers. **a**, **b** Maximum Z-projects (**a**) and single slice from Z-stack (**b**) corresponding to confocal overlay images of double-immunofluorescence labeling for FXR1P (green) and α-actinin, phalloidin, RYR and COX-IV (red) in isolated EDL fibers from different mice. Scale bars 5 μm. Arrowheads in panel **a** point to Z-lines, and in panel **b** (delACAG COX-IV/FXR1P image) indicate co-localization of FXR1P granules and mitochondria. Nuclei are stained with DAPI (blue). **c** Representative RGB intensity profile along a line corresponding to a Z-stack slice of a wt EDL fiber showing co-localization results between FXR1P (green) and sarcomeric markers (red). FXR1P is adjacent to Z-lines and mitochondria. Experiments were performed in n = 3 mice except for RYR (n = 2 mice). For each mouse several fibers were analyzed in each experiment

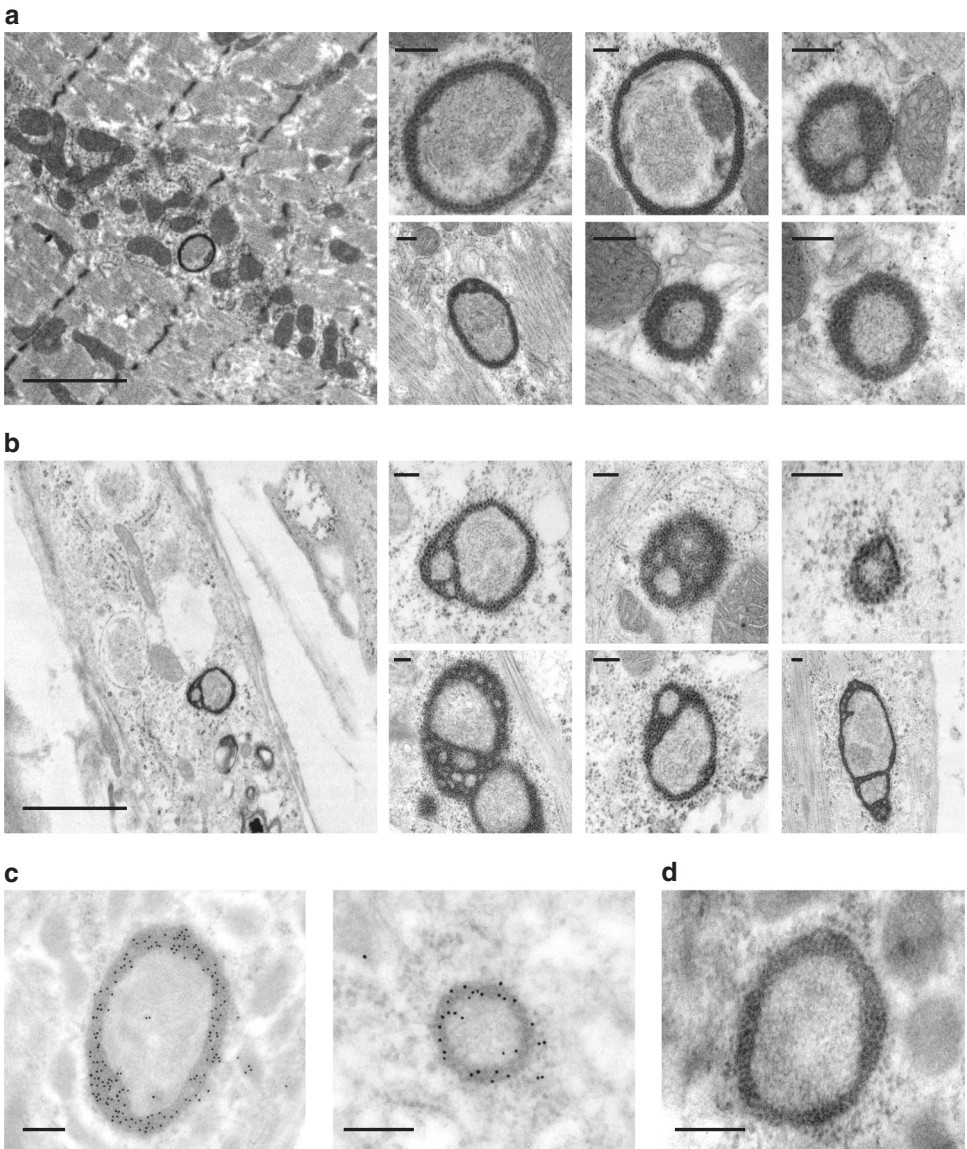

**Fig. 7** Ultrastructural evaluation of delACAG granules. **a** Representative TEM images of granules of different size exclusively observed in gastrocnemius fibers from delACAG mice. Granules are often located in close proximity to mitochondria. $n = 3$(wt), 4(delACAG), 2(dupA) mice. Scale bars 2 µm (left panel), 200 nm (rest of panels). **b** Representative TEM images of ring-shaped granules from V-4 differentiated myotubes. These structures were not present in control differentiated myotubes. Multiple myotubes were analyzed. Scale bars 2 µm (left panel), 200 nm (rest of panels). **c** Anti-FXR1P immunogold labeling of ultrathin sections from gastrocnemius of delACAG mice showing localization of the mutant protein in the outer electrodense layer of delACAG granules. $n = 3$. Scale bar 200 nm. **d** No-primary antibody control demonstrating no immunogold labeling in granules from delACAG mice. Scale bar 200 nm

The phenotype of exon-15 mutant mice indicate that the dupA mutation is less severe than the delACAG variant which correlates with the different effect of each mutation since: (1) the amount of FXR1P at Z-lines was significantly lower in delACAG than in dupA fibers, and (2) only delACAG isoforms accumulated in distinctive granular structures which interact with mRNA. Thus, the pathology of delACAG mice not only results from absence of P82,84 from their normal subcellular localization, but also from their accumulation in granules which, in addition to disturb the structure of the fiber, interfere with mRNA trafficking.

Gene expression profiling of delACAG muscle tissue identified p53 signaling as a main GO/KEGG pathway and revealed induction of genes related to cell cycle arrest, DNA damage response, apoptosis and inflammation some of which are p53 targets. The increased levels of *Trp73* and *Trp63* transcripts in

delACAG mutants are potentially responsible for p53 pathway activation in these mice as they share functional roles with p53[28,29]. *Cdkn1a* overexpression in delACAG mutants could then result from p73 induction, or from FXR1P loss-of function since *Cdkn1a* mRNA is destabilized by FXR1P[26,30,31]. *Bex1* encodes a ribonucleoprotein that mediates mRNA stabilization of proinflammatory genes including *Cxcl10*[32]. The transcript levels of both *Bex1* and *Cxcl10* were found increased in delACAG mice, thus pointing to deregulation of the pro-inflammatory program in these mice. Notably, a role of FXR1P in controlling abundance of pro-inflammatory transcripts in vascular smooth muscle cells has recently been reported[33]. Several delACAG DEGs are common to other myopathies. These genes are related to fiber regeneration, such as *Myh3* or to muscle injury and atrophy like *Ankrd1*, *Gdf15*, *Mstn* or *Nos1*[34–36]. DelACAG mice were also

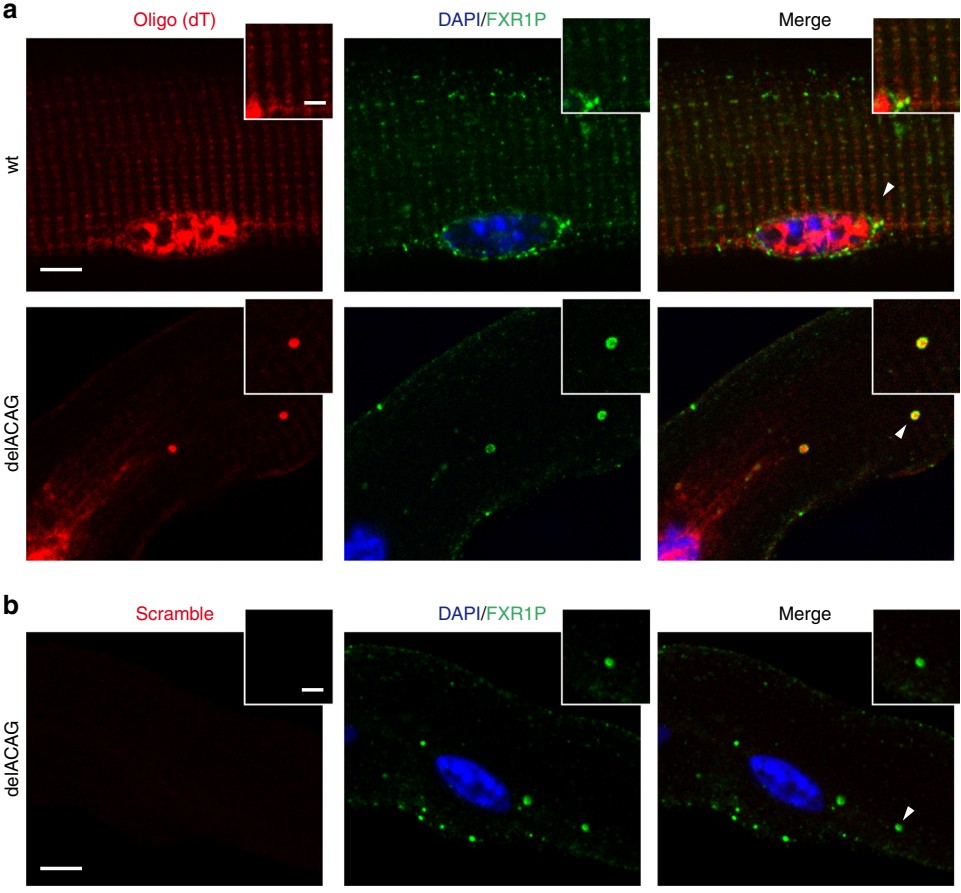

**Fig. 8** DelACAG granules incorporate mRNA. **a**, **b** Confocal images (single Z-slice) of RNA-FISH on EDL isolated myofibers (wt and delACAG) using Cy3-oligo(dT) (**a**) or a Cy3-scramble probe (**b**) in combination with anti-FXR1P immunostaining proving positive oligo (dT) signal in delACAG granules ($n = 7$). Scale bar 5 μm. Higher magnifications correspond to the areas designated by arrowheads. Scale bar 2 μm. Nuclei are stained with DAPI

identified with high levels of both sarcolipin mRNA and protein. This suggests abnormal $Ca^{+2}$ handling associated with mutations in P82,84, since SLN is an inhibitor of SERCA channels, which pump $Ca^{+2}$ back from the cytosol to the SR after excitation-contraction[37]. Concordantly, approximately 50% of patients with multi-minicore myopathies are found with recessive mutations in *RYR1* and *SEPN1* which are also genes related to $Ca^{+2}$ homeostasis[38,39]. RYR1 is a $Ca^{+2}$ channel that allows release of $Ca^{+2}$ from the SR and SEPN1 regulates RYR1[40]. Additional genes have been implicated in minicore myopathy. Recessive mutations in *TTN*, *ACADS*, *MEGF10*, and *SECISBP2*, dominant mutations in *ACTA1* and *MYH7*, and one mutation in *CCDC78* in a single family with dominant inheritance have been described in patients with multi-minicore disease[39].

In conclusion, we identified five individuals with neuromuscular phenotypes from two unrelated families with different recessive mutations in the same exon of *FXR1*. DelACAG mice reproduced the effect of the human mutation on the protein and demonstrated pathologic features very similar to those of family 2 patients, specifically muscle atrophy, multicore lesions, central nuclei and type I fiber predominance. Although patients from family 2 had neonatal hypotonia and delayed gross motor milestones, the clinical outcome of all three siblings was milder compared with that observed in patients of family 1. The phenotypic differences between delACAG and dupA mice provide support for the phenotypic variability among patients with different *FXR1* mutations. Indeed, similar to dupA mice, which have milder phenotype than delACAG mutants, no granular accumulations of FXR1P were distinguished in family 2 by TEM

analysis of muscle tissue. Regardless, in addition to possible differences in P82,84 expression levels and/or localization between both families, variable expressivity in monogenic neuromuscular disorders is common, as seen in the range of phenotypes associated with *RYR1* mutations[41] and among patients with spinal muscular atrophy[42]. Considering cardiac muscle, the proband of family 1 had episodes of tachycardia, but no apparent myocardial dysfunction was detected in members of family 2. However, since P82,84 are the main FXR1P isoforms in cardiac muscle, a potential effect of *FXR1*-exon-15 mutations in this tissue cannot be disregarded. The overexpression of proinflammatory pathways in delACAG mice could contribute to the bone phenotype observed in these mice since pro-inflammatory cytokines have adverse effects on BMD[43]. However, decreased fetal movement, which often leads to arthrogryposis, hypomineralized bones and fractures[44], is the most likely cause of the bone and arthrogryptotic phenotype of family 1. The full phenotypic spectrum associated with *FXR1* mutations will continue to expand as more patients with *FXR1*-related disorders are recognized.

## Methods

**Human samples**. All patients or patient's guardians provided written informed consent to participate in this study and for publication of clinical information, molecular findings and photographs (Fig. 1). All studies and investigations were approved by the Medical Research Ethics Committees of the National Research Centre (Egypt), Hospital La Paz and Consejo Superior de Investigaciones Científicas (CSIC) and by the Research Ethics Board of the Hospital for Sick Children (REB #1000009004). Tissue samples to conduct research were obtained with appropriate informed consent according to the declaration of Helsinki principles of medical research involving human subjects.

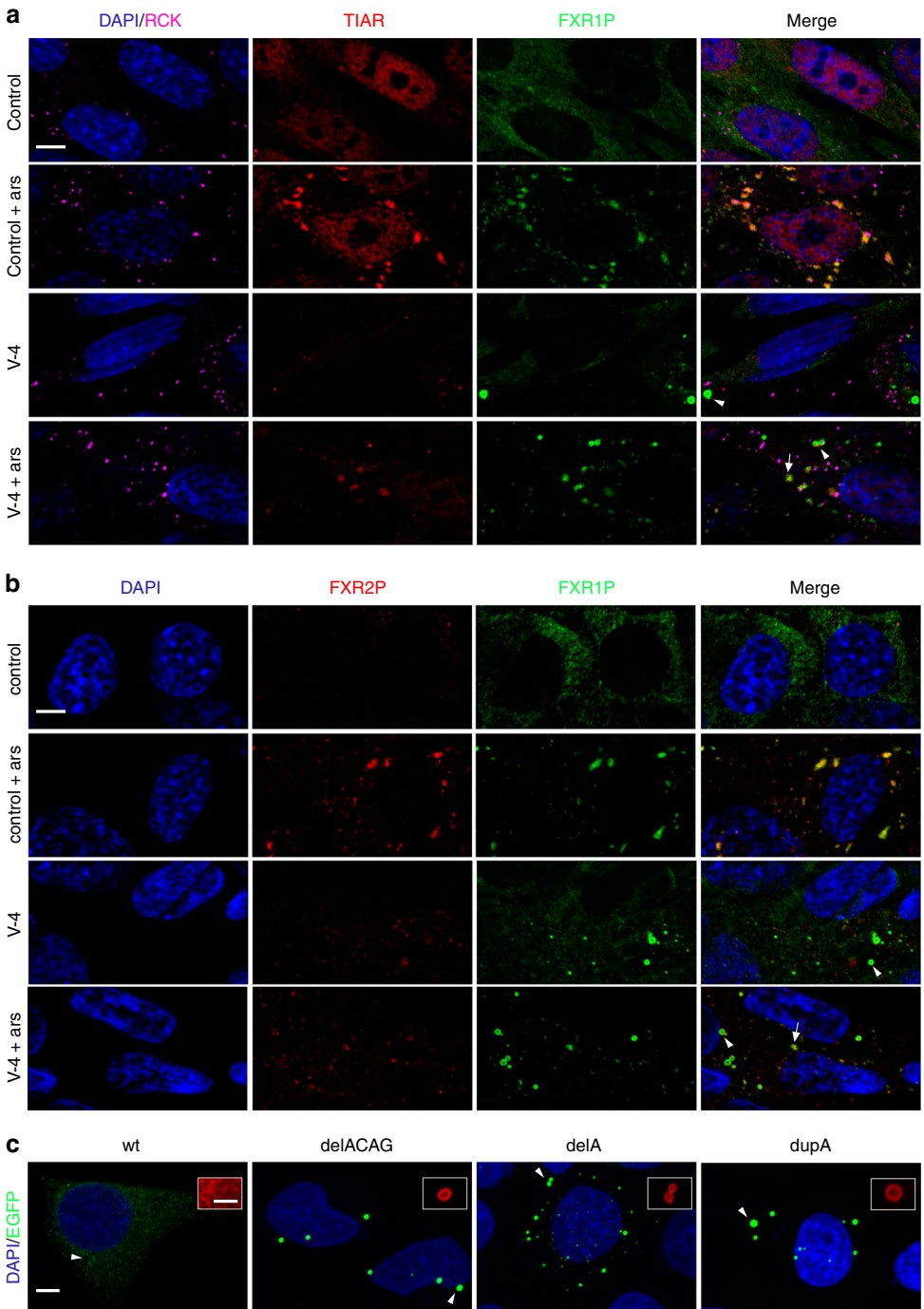

**Fig. 9** DelACAG granules do not recruit SG components. **a, b** Representative images of confocal microscopy (single Z-slice) from one-day differentiated V-4 or control myoblasts treated with or without ARS and co-immunostained for FXR1P and TIAR/RCK or FXR2P. Strong co-localization of FXR1P with TIAR and FXR2P occurs in amorphous SG granules (arrows), but not in ring-shaped delACAG granules (arrowheads). n = 4(TIAR/RCK), n = 3(FXR2P). Scale bars 5 μm. **c** Confocal (Z-project) representative images of transfected HeLa cells with EGFP:FXR1P-iso-e fusion proteins (Supplementary Fig. 10a) illustrating assembly of the three mutant proteins (delACAG, delA, and dupA) into ring-shaped delACAG-like granules. Equivalent images of transfected HeLa cells labeled with anti-FXR1P are in Supplementary Fig. 10c. Scale bar 5 μm. Higher magnifications correspond to anti-FXR1P staining of the areas designated by arrowheads. n = 3. Scale bar 2 μm. In all panels nuclei are stained with DAPI

**Genetic analysis**. Homozygosity mapping in the proband of family 1 was conducted by whole-genome SNP-array hybridization using CytoSNP-850k BeadChip SNP-based arrays (Illumina) as indicated in Palencia-Campos et al[45]. For this proband WES was provided by Sistemas Genomicos S.L. (Valencia, Spain). Targeted regions were captured using SureSelect Human All Exon Target Enrichment kit for 51 Mb (Agilent Technologies) and reads were aligned against human genome assembly GRCh37/hg19. Initial filtering was performed with Picard-tools (http://broadinstitute.github.io/picard/) and SAM-tools[46] and variant calling was obtained using combination of VarScan[47] and GATK[48]. Identified variants were annotated using Ensembl database (www.ensembl.org). The filtering pipeline is in Supplementary Table 1a. SNP-array hybridization in V-4 was performed as in the proband after biological material became available. Multiplex Ligation-dependent Probe Amplification (MLPA) of the SMN locus was accomplished using SALSA probe mix P021 (MRC-Holland) according to the manufacturer specifications. For WES in family 2 individuals, 2 × 100 bp paired end sequencing was carried out using the Illumina Hi-Seq 2000 platform after target enrichment of 6.5 μg of genomic DNA with the Agilent SureSelect Human All Exon 50 Mb Capture kit. Overall mean exon coverage was 177–190X with ≥20X base coverage of 96.7-97.6%.

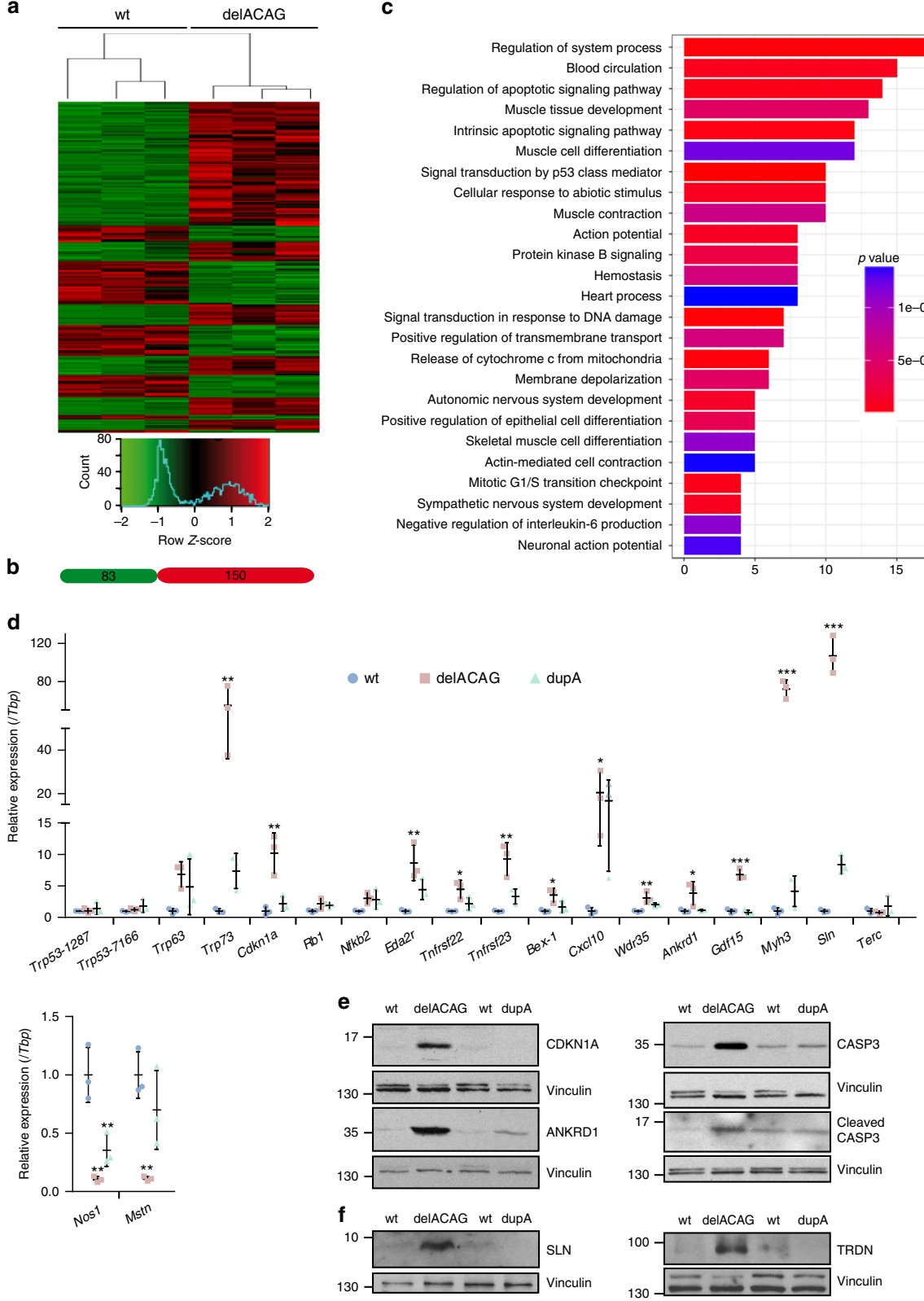

Sequence reads were aligned to the reference human genome (hg19/GRCh 37) with Burrows-Wheeler aligner[49] and paired-end duplicate reads were removed (Mark-Duplicates, Picard tools v.1.35; http://picard.sourceforge.net/). Single nucleotide variants (SNVs) and insertions/deletions (indel) were detected using GATK 1.1.28[50] and annotated using SNPEff (http://snpeff.sourceforge.net/). Data was analyzed using seqr. Variants were also annotated for frequency in public databases [1000 genomes; dbSNP135, http://www.ncbi.nlm.nih.gov/projects/SNP/; NHLB1 Exome Sequencing Project (ESP) Exome Variant Server, http://evs.gs.washington.

edu/EVS/]. The potential effect of SNVs were predicted using Polyphen and SIFT. Array based comparative genomic hybridization (aCGH) in patients from family 2 was conducted in DNA isolated from lymphoblast cell lines (LCL) using an in-house-designed 8 × 60k genome-wide oligonucleotide custom array with an average probe spacing of 43 kb (KaryoArray® v3.0; Agilent-based; Agilent Technologies). Hybridization experiments were performed as recommended by the manufacturer (Agilent Technologies). Analysis and visualization of Karyoarray® was achieved with Genomic Workbench Standard Edition 5.0 software (Agilent Technologies).

**Fig. 10** Gene expression profiling in skeletal muscle from delACAG mutants. **a** Hierarchical clustering and heatmap visualization of RNA-seq data (selected DEGs) resulting from analysis of skeletal muscle RNA samples isolated from 3 wt and 3 delACAG P15 mice. **b** Number of genes upregulated (red) and downregulated (green) among selected 233 DEGs. **c** Gene Ontology (GO) term enrichment analysis showing pathways and biological processes over-represented in selected DEGs. Gene count is shown on the X-axis (Supplementary Table 3a). **d** Relative expression analysis of the indicated genes by qRT-PCR using TaqMan expression assays in gastrocnemius RNA of 1.5-months old mice. Values are normalized to *Tbp* expression and are represented as fold change of the mean value of wt expression. Graphs are mean ± s.d. followed by one-way ANOVA with Tukey post-hoc test. $n = 3$. **e** Representative immunoblots showing protein levels of CDKN1A, ANKRD1, CASP3 and cleaved CASP3 in gastrocnemius from 2–3-months old delACAG and dupA mice and corresponding wt littermates. **f** Representative immunoblots for SLN and TRDN in protein extracts from the soleus (SLN) or gastrocnemius (TRDN) from delACAG, dupA and wt mice demonstrating increased expression of both proteins in delACAG mutants. Vinculin served as loading control in **e** and **f**. $n = 6$(wt), 3(delACAG), 3(dupA) mice for all proteins excepting SLN ($n = 7$(wt), 4(delACAG), 3(dupA) mice)

Whole-genome SNP-arrays in family 2 were carried out as in family 1 using LCL genomic DNA.

**Cell culture**. V-4 primary myoblasts were isolated from a muscle biopsy (quad-riceps femoris, medial head) and cultured according to a published protocol[51]. Briefly, the muscle tissue was mechanically fragmented and incubated at 37 °C/5% $CO_2$ in 0.2% colagenase I (Sigma)/DMEM for 1 h followed by another incubation in 0.05% trypsin (Gibco)/ HBSS(Gibco) for 30 min. The digested sample was filtered through a 100 μm nylon cell strainer and plated in 0.2% collagen pre-coated dishes in growth medium (Ham's F10 + 20% FBS + 1% penicillin/streptomycin (P/S), (Gibco)). In all experiments primary myoblasts from 4–7 passages were used. Differentiated myotubes were obtained by starving confluent myoblasts (90%) in differentiation medium (Ham's F10 + 2% horse serum (HS, Gibco) + 1% P/S) for 5 days. Human primary fibroblasts were generated from skin biopsy after digesting the tissue in 0.05% trypsin/PBS for 30 min at 37 °C/5% $CO_2$ and cultured in DMEM + 10% FBS + 1% P/S. Mouse primary myoblasts were obtained from lower limb muscle packages of 2–3 months old mice. Mouse cells were isolated and cultured as described for human myoblasts except that they were subjected to several pre-plating steps in non-coated dishes to ensure fibroblast removal. HeLa and HEK293T were cultured in DMEM + 10% FBS + 1% P/S. For SG induction, myoblasts and transfected HeLa were incubated in 0.5 mM of arsenite (ARS, Sigma) diluted in the corresponding cell culture media for 45 min before fixation. For HeLa, the media was refreshed 1 h prior ARS incubation.

**Directed mutagenesis, plasmids and transfections**. The cDNA of wt FXR1 iso-e was generated by standard RT-PCR from human muscle mRNA purchased to Clontech, cloned into pBluescript SK(+) and sequenced. A plasmid containing wt iso-e sequence (XM_005247813.3) was used as template to create the delACAG, delA and dupA mutations using the QuikChange II XL Site-Directed Mutagenesis Kit (Agilent Technologies). Wt and mutant iso-e variants were subsequently transferred into pEGFP-C1 (Clontech) to generate the corresponding EGFP: FXR1P-isoe fusion proteins. All constructs were verified by sequencing. HeLa seeded onto coverslips in 24-well plates were transfected with 500 ng of DNA/well using Lipofectamine 3000 (Invitrogen) and OptiMEM medium (Gibco). After 5–7 h the media was replaced with DMEM + 10% FBS + 1% P/S and 48 h after transfection, cells were ARS-treated and fixed. HEK293T were transfected by standard calcium phosphate method and used to corroborate expression of EGFP: FXR1P-iso-e fusion proteins by WB. HeLa and HEK293T tested negative for mycoplasma.

**Transgenic mice**. Fxr1 edited mouse founders were generated by genOway (Lyon, France). A CRISPR sgRNA specifically designed to target Fxr1 exon15 and generate the 4-bp ACAG deletion was microinjected into fertilized oocytes. Newborn mice were screened by PCR and sequence analyzed to assess Fxr1 exon-15 sequence. Three founders were identified and bred to obtain heterozygous animals. Three mouse lines were generated, two harboring the 4-bp deletion (delACAG) and one harboring the adenine duplication (dupA). CRISPR mouse lines were maintained independently in a C57BL/6 J genetic background by crossing heterozygous males to wt females. Heterozygous offspring from these crosses were mated to generate homozygous mutants and control mice for experimental purposes. Homozygous mutants from both independent delACAG lines were analyzed and confirmed to produce identical results. Experiments were performed using homozygous mutants and control wt mice offspring of F2 to F5 heterozygous crosses without finding any difference in phenotypes. Studies with conditional knockout mice were carried out in a C57BL/6 J background. Fxr1 floxed mice[13] and Myf5-Cre transgenics[16] were previously reported. Tg.CAG-Cre deleter mice were used to achieve constitutive global excision of Fxr1 exon-1. Experiments involving mice were performed according to ethical regulations and were approved by the corresponding Ethics committees of Instituto de Investigaciones Biomédicas "Alberto Sols" and CSIC.

**Histopathology**. Vastus lateralis, gastrocnemius and soleus of 3 month old female mice were dissected, embedded together in OCT, and frozen by submersing in isopentane pre-cooled in liquid nitrogen. Transverse cryostat sections (10 μm)

containing the three muscles were stained with haematoxylin-eosin (H-E) or used for NADH-TR and ATPase 4.3 enzymatic histochemistry following standard procedures. Pictures were captured with a Zeiss Axiophot microscope. H-E cross-sections of vastus were used to quantify cell number/microscope field of view ($375.56 \times 282.78$ μm²), cell size (minimal Feret's diameter: MFD) and the percentage of fibers with central nuclei. For each mouse, 3 independent images from the same cryosection were counted. Relative frequency of MFD (%) was determined using length intervals of 5 μm. Quantification of type 1 fibers using ATPase (pH 4.3) histochemistry was conducted in an entire cross-section of soleus. For histo-pathological analysis of family 2, a muscle biopsy of the right triceps was performed from II-4. Biopsy portions were fixed in formalin for light microscopy, and snap frozen for histochemistry and immunohistochemistry. Using standard histo-chemical protocols, cryostat sections were processed for H-E, modified Gomori trichrome, Masson trichrome staining, Periodic acid Schiff, oxidative enzymes, Oil red O, Congo Red, and ATPase at pH 9.4 and 4.3. Samples for TEM were fixed and processed according to standard protocol.

**Electron microscopy of mouse tissue and differentiated myotubes**. Gastroc-nemius specimens of 3 month old female mice were fixed in 1% glutaraldehyde/ 4% PFA in 0.1 M cacodilate buffer overnight, postfixed in 1% osmium tetroxide in water for 60 min and dehydrated through a series of ethanol solutions and acetone. After the last dehydration step, samples were incubated in a 1:3, 1:1, 3:1 mixture of durcupan resin and acetone and cured at 60 °C for 48 h. Longitudinal ultrathin sections (50–60 nm) were obtained with a diamond knife (Diatome) in an ultra-microtome (Leica Reichert ultracut S) and collected in 200-mesh copper grids. Sections were counterstained with 2% uranyl acetate in water for 20 min followed by a 10 min incubation step with a lead citrate solution. Samples were examined with a JEOL JEM1010 electron microscope (Jeol Ltd., Tokyo, Japan). For ultra-structural analysis of cultured myotubes, myoblasts were seeded and differentiated in Permanox plastic chamber slides (Lab-Tek) and subsequently fixed and embedded as described above except that acetone was not included in the dehy-dration steps. Myotubes were embedded in Epon resin, sectioned and counter-stained as indicated for muscle tissue.

**Immunogold labeling**. Immunogold staining was performed in samples from gastrocnemius of delACAG and control mice fixed in 0.25% glutaraldehyde/4% PFA in phosphate buffer 0.1 M, dehydrated and embedded in LR White resin. Ultrathin sections of 60 nm collected in 200-mesh Ni grids were blocked and incubated with anti-FXR1P (Proteintech; 1:100) and corresponding secondary antibody (Aurion, 10 nm gold conjugate goat anti-rabbit; 1:30) according to the post-embedding immunogold protocol recommended by the manufacturer (Aur-ion). Samples were postfixed in 2% glutaraldehyde/PBS and contrasted with 2% uranyl acetate/$H_2$0.

**Magnetic resonance imaging (MRI)**. MRI examination of mice was performed in 14 weeks old female mice in a 7.0 T 16 cm bore Bruker Biospect (Bruker Medical Gmbh, Ettlingen, Germany). Mice were maintained anesthetized with 1–2% iso-flurane in 1 L of oxygen through the experiment. T2-weighted spin echo anato-mical images were acquired with a rapid acquisition with relaxation enhancement sequence in axial and coronal orientations with the following parameters: TR = 3000 ms, TE = 11.57 ms, RARE factor = 8, Av = 12, FOV = 2.5 cm, acquisition matrix = $256 \times 256$ corresponding to an in plane resolution of $97 \times 97$μm², slice thickness = 1 mm without gap between slices. To calculate muscle volume, the muscular area was measured in 5 consecutive slices of hindlimbs, starting at 2 slices after tibial plateau in each mouse. Muscle volume value was the mean of both hindlimbs. Human muscle MRI for II-5 (family 2) was performed of both upper and lower proximal muscles utilizing axial and coronal Short-TI Inversion Recovery (STIR) and T1-weighted images.

**Dual-energy x-ray absorptiometry (DXA) and X-rays**. Bone mineral density (BMD, g·cm$^{-2}$) was assessed in 4-month old anesthetized female mice by dual-energy X-ray absorptiometry using PIXImus (GE Lunar Corp., Madison, WI, USA) as previously explained[52]. This parameter was measured in the selected ROI

(region of interest, cm$^2$) in vertebrae L4–L5 (35 × 23 pixels) and femur (11 × 33 and 17 × 13 pixels), defining trabecular and cortical bone values, respectively. For X-rays fore-limbs and hind-limbs of 3.5 or 4-months female mice were dissected and stripped of muscle tissue prior to be analyzed. In each independent experiment the limbs of all genotypes were X-rayed simultaneously on the same film using a HP Faxitron 43855A instrument with 12 KV intensity for 10 s. Films were subsequently photographed. Mice of the same age were used in each experiment.

**Functional performance tests.** Muscular strength and endurance were assessed in 14-weeks-old female mice using three different tests: rotarod, weight test and hanging wire test. Rotarod fatigue assay was conducted using a 47600 Rotarod Instrument (Ugo Basile, Italy) following the protocol published by Carrell EM et al.[53]. For the weight test we created 5 different weights, consisting of a 7 g ball of metallic thread (0.33 mm thickness) to which we attached a 1–5 steel chain links each weighting 12 g. Final weights: 19, 31, 43, 55, and 67 g. The weight test was carried out as explained by Deacon RM[54] and was repeated three consecutive days in each mouse. Scores were calculated as mean ± s.d. per animal. The hanging wire test was performed according to standardized operating procedure described by Maaike van Putten et al. (TREAT-NMD, Neuromuscular Network, http://www.treat-nmd.eu/resources/research-sops/dmd-sops/, 2016) using the falls and reaches-method. In addition to the falling and reaching scores we introduced a new score that we named "four limbs" based on the number of times that the mouse was able to lift both hindlimbs and grabbed the wire with all four extremities. This score starts from zero and increases by 1 each time that an event of this type occurs in a total of 180 s and until a maximum score of 10.

**Immunoblotting.** Muscle tissues were homogenized using a T10 basic ULTRA-Turrax (IKA) in RIPA lysis buffer (150 mM NaCl, 50 mM TrisHCl, 2 mM EDTA, 0.5% Sodium Deoxicolate, 0.1% SDS, 1% NP-40) supplemented with protease inhibitors (Sigma P8340, 1 mM PMSF) and phosphatase inhibitors (Sigma P0044 and P5726, 1 mM Na$_3$O$_4$V and 10 mM NaF). Following a 20 min incubation step at 4 °C, the tissue homogenate was sonicated in a cold bath at max power for 1 min, and clarified by centrifugation at 4 °C (15 min, 14000×$g$). Supernatants were collected and considered as the soluble fraction (SF), and the pellets (insoluble fraction, IF) were re-dissolved for 30 min in 8 M urea in the presence of protease and phosphatase inhibitors and subsequently sonicated (3 cycles of 1 min). Protein quantification was conducted with BCA Protein Assay (Pierce). Protein samples (40 μg) were separated by SDS-PAGE, transferred onto nitrocellulose membranes and subjected to WB. Embryonic tissue homogenates and cultured cell lysates were obtained in RIPA buffer following the procedure indicated for muscle tissue. The IF fraction of cultured myoblasts and myotubes was extracted in urea, as explained above. Primary antibodies: FXR1P all isoforms (Proteintech, 13194-1-AP, 1:4000; #ML13[55] serum 1:10000), FXR1P iso-e/f-specific (#27-15 and #27-17 policlonal sera, 1:4000)[5], α-actinin (Millipore 05-384, 1:4000), MHC (R&D systems MAB4470, 1:10000), α-tubulin (Sigma T9026, 1:20000), vinculin (Santa Cruz sc-73614, 1:1000), caspase-3 (Cell Signaling #9665, 1:1000,), cleaved (active) caspase-3 (Cell Signaling #9661, 1:500), CDK1NA (Santa Cruz sc-397, 1:500), ANKRD1 (Proteintech 11427-1-AP, 1:1000), triadin (Thermofisher MA3-927, 1:1000), SLN (1:100)[56], and GFP (Santa Cruz sc-9996, 1:1000). HRP-conjugated secondary antibodies were from Jackson ImmunoResearch. WB were developed using ECL reagent (Amersham). Anti-FXR1P antibodies Proteintech 13194-1-AP and #27-15 were demonstrated to recognize FXR1P isoforms using cells transfected with EGFP:FXR1-iso-e isoforms (Supplementary Fig. 10). Uncropped scans of WBs are presented in Supplementary Fig. 13.

**Isolation of EDL fibers and confocal microscopy.** Dissected EDL muscles were digested in 0.2% collagenase type I (Sigma)/DMEM for 1 h at 37 °C/5% CO$_2$ and subsequently transferred to HS pre-coated Petri dishes containing fresh DMEM + 1% P/S to release myofibers with the help of gentle flushing as described by Pasut et al.[57]. For immunoanalysis, isolated EDL fibers were plated in chamber slides (Lab-Tek) precoated with 3% matrigel and incubated in DMEM + 2%HS + 1% P/S for 2 h before being fixed in pre-warmed 4% PFA/PBS for 5 min at room temperature. Fibers were next permeabilized in 0.5% Triton X-100/PBS for 10 min, blocked in 4% goat serum/0.1% Triton X-100/PBS for 30 min and incubated with primary antibodies diluted in blocking solution overnight at 4 °C. After several washes with 0.1% Triton X-100/PBS the corresponding secondary antibodies (Molecular Probes) and/or Phalloidin-546 (Molecular Probes A22283, 1:200) were applied for 1 h at room temperature. Nuclei were stained with DAPI (1:1000 in PBS) or propidium iodide (Sigma-Aldrich; 20 μg·ml$^{-1}$ in 0.05% Tween-PBS). Chamber slides were mounted with Prolong Diamond. Myoblasts, differentiated myotubes and HeLa were grown in coverslips, fixed in 4% PFA/PBS for 10 min, permeabilized with 0.1% Triton X-100/PBS for 15 min and blocked in 4% serum/PBS-0.05% Triton X-100 for 30 min. Cells were incubated with primary antibody overnight 4 °C, washed 3 times with 0.05% Tween/PBS and then incubated with fluorescently labeled secondary antibodies (Molecular Probes) for 1 h, washed and DAPI stained before being mounted with Prolong Diamond. Primary antibodies: FXR1P (Proteintech 13194-1-AP, 1:200), α-actinin (Millipore 05-384, 1:200), RYR (Abcam ab2868, 1:200), COX-IV (Invitrogen 459600, 1:200), desmin (Abcam ab6322, 1:100), MHC (R&D MAB4470, 1:2000), TIAR (Santa Cruz sc-1749, 1:30),

FXR2P (Santa Cruz sc-32266 1:60), FMR1P (DSHB 7G1-1, 1:70), RCK (Santa Cruz sc-376433, 1:30). Fluorescence images were acquired with a Leica SP5 confocal microscope or Zeiss LSM710.

**RNA-FISH.** Isolated EDL fibers were fixed in 4% PFA (10 min) and permeabilized in 0.5% Triton X-100/PBS (30 min). Subsequently, fibers were incubated in pre-hybridization buffer (30% formamide, 2xSSC) for 10 min at room temperature and hybridized for 2 h at 37 °C with 1 ng·μl$^{-1}$ of Cy3-labeled oligonucleotide probe in hybridization buffer (30% formamide, 10% dextran sulfate, 1 mg·ml$^{-1}$ yeast tRNA, 2xSSC, 2 mM vanadyl complex, 0.005% BSA). Poly(A)$^+$ mRNAs were detected using a 5′-Cy3 labeled oligo-(dT)$_{30}$, and a previously published 5′-Cy3-tagged scrambled oligonucleotide (5′-TGTAACACGTCTATACGCCCA-3′)[19] was used as negative control. Following hybridization, fibers were washed in 30% formamide 4x SSC (20 min), 4xSSC (20 min), 2xSSC (20 min) before immunolabelling. Anti-FXR1P (Proteintech 1:200) and secondary antibody (1:1000) were diluted in 0.1% Triton X-100/2xSSC and subsequent washes were performed in 2xSSC. Fibers were stained with DAPI, mounted with Prolong and evaluated by confocal microscopy. The same procedure was applied for human cultured differentiated myotubes.

**Transcriptomics.** For RNAseq, 6 RNA samples corresponding to 3 wt and 3 delACAG mice were used. RNA samples were isolated from gastrocnemius and soleus of P15 mice using TriReagent solution (Ambion) according to the manufacturer protocol. Library generation and RNA sequencing were performed at Sistemas Genómicos S.L. (Valencia, Spain). Ribosomal RNA was removed with the Ribo-Zero rRNA removal kit (Illumina) and generation of index-tagged cDNA libraries was performed with the TruSeq Stranded Total RNA library Prep kit from 2 μg of total RNA. Sequencing was conducted in a HiSeq2500 instrument (Illumina) and sequencing readings were paired-end with a length of 101 bp. Estimated coverage was around 52 million reads per sample. Quality control of the raw data was assessed using the FastQC v0.11.4 tool and the raw paired-end reads were mapped to the mouse genome provided by Ensembl database (GRCm38) using Tophat2 2.1.0 algorithm[58]. Insufficient quality reads (phred score <5) were eliminated using Samtools 1.2[46] and Picard Tools 2.12.1. The differential expression analysis between wt and delACAG samples was conducted using statistical packages designed by Python and R. Expression levels were calculated using the HTSeq[59] and differential expression analysis between conditions was assessed using DESeq[60] applying a differential negative binomial distribution for the statistics significance[61]. We selected differentially expressed genes with a $P$ value adjusted by FDR[61] ≤ 0.05 and fold change value below −1.9 or higher than 1.9 to avoid identification of false positives across the differential expression data. For Gene Set functional enrichment analysis and network analysis, differentially expressed sets were processed using ClusterProfiler[62], a Bioconductor package to search for biological processes involved. This tool screens for genes in specific databases (Gene Ontology (GO), Kyoto Encyclopedia of Genes and Genomes (KEGG)) to evaluate biological annotations that rise as over-represented with respect to the whole genome.

**qRT-PCR.** Gastrocnemius from age-matched normal and mutant mice were dissected and maintained in RNAlater (Ambion) until processing. The tissue was homogenized and RNA extracted with TriReagent solution according to the manufacturer's protocol. cDNA was synthesized from 250 ng of total RNA with High Capacity cDNA Reverse Transcription Kit (Applied Biosystems) using random primers. Quantitative RT-PCR was performed in a 7900HT Fast Real-Time PCR System (Applied Biosystems) using TaqMan real-time PCR gene expression assays (Applied Biosystems). For each TaqMan assay three mice per genotype were analyzed and every sample was run in triplicates. Transcripts levels were normalized against expression of housekeeping genes (*Tbp*, *Gusb*, *β2M*). Fold differences were calculated by the 2$^{-ΔΔCt}$ method using as calibrator sample the mean value of wt mice. The following assays were used: *Fxr1* (exons 2–3; Mm00484523_m1), *Fxr1* (exons 14–15; Mm01286304_m1), *Trp53* (exons 1–2; Mm01731287_m1), *Trp53* (exons 9–10 Mm01337166_mH), *Trp63* (Mm00495788_m1), *Trp73* (Mm00660220_m1), *4930567H12Rik* (*Cdkn1a*, Mm00432448_m1), *Rb1* (Mm01310562_m1), *Nfkb2* (Mm00479807_m1), *Eda2r* (Mm01236761_m1), *Tnfrsf22* (Mm00445826_m1), *Tnfrsf23* (Mm00656375_m1), *Bex1* (Mm00784371_s1), *Cxcl10* (Mm00445235_m1), *Wdr35* (Mm00552654_m1), *Ankrd1* (Mm00496512_m1), *Gdf15* (Mm00442228_m1), *Myh3* (Mm01332463_m1), *Sln* (Mm00481536_m1), *Terc* (Mm01261365_s1), *Nos1* (Mm01208059_m1), *Mstn* (Mm01254559_m1), *Tbp* (Mm00446973_m1), *Gusb* (Mm01197698_m1) and *β2M* (Mm00437762_m1). For TaqMan assays capable to detect DNA (*Cdkn1a*, *Bex1*, and *Terc*), RNA samples were treated with Turbo-DNA-free Kit (ThermoFisher) to eliminate possible residual DNA contamination. For *Fxr1* expression analysis, 2.5 month old mice were used and for the rest of genes RNA was obtained from mice of 1.5 months of age.

**Image and statistical analyses.** Statistical analysis was conducted with GraphPad software. D'Agostino-Pearson, Shapiro-Wilk, Kolmogorov-Smirnov normality tests or Skewness and kurtosis normality score were used to assess normal distribution of data. Variance F-test, Brown-Forsythe or Bartlett's tests were used to test homoscedasticity of variance. Parametric tests including one-way ANOVA with

Tukey's multiple comparisons test or two-tailed unpaired Student's *t*-test were used after validation of normality and homoscedasticity of variance. When normality was not assumed non-parametric Kruskal-Wallis with Dunn's multiple comparison test as post-hoc was used. No randomization, predetermination of sample size or blinding was used. *P*-values ≥ 0.05 were considered statistically non-significant; ***$P < 0.001$; **$P < 0.01$; *$P < 0.05$. Image analysis, quantifications and WB densitometry were conducted with Image J-Fiji software.

## Data availability

The RNA-Seq data from this study has been deposited in NCBI Sequence Read Archive (SRA) database with accession number SRP150400.

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

## Acknowledgements

We are grateful to patients and their families for their participation in this study. We thank Dr. Edward W Khandjian for the kind gift of anti-FXR1P antibodies (#ML13, #27-15, and #27-17)[5,9,12,55], Dr. Robert Bloch for the anti-SLN antibody[56] and the Developmental Studies Hybridoma Bank, created by the NICHD of the NIH and maintained at The University of Iowa, for monoclonal antibodies. We also would like to thank M. Dolores García-Concepción, M. Gracia González-Bueno, Francisco R Urbano, Covadonga Aguado, José A Rodríguez, Mónica Martín-Belinchón, Diego Navarro, the Genomics and MRISS core facilities at IIB, Carmen Sánchez-Palomo, Rupleen Kaur, Juan C Triviño, and Guillermo Marco for technical assistance. This work was financially supported by the Spanish Ministry of Economy and Competitiveness (SAF2013-43365-R/SAF2016-75434-R) and CIBERER (ACCI 2017). The work performed at the NIH was supported by intramural funds from the NIH National Institute of Neurological Disorders and Stroke. Sequencing analysis was provided by the Broad Institute of MIT and Harvard Center for Mendelian Genomics and was funded by the National Human Genome Research Institute, the National Eye Institute and the National Heart, Lung and Blood Institute grant UM1 HG008900 to Daniel MacArthur and Heidi Rehm. The Broad Center for Mendelian Genomics (UM1 HG008900) is funded by the National Human Genome Research Institute with supplemental funding provided by the National Heart, Lung, and Blood Institute under the Trans-Omics for Precision Medicine (TOPMed) program and the National Eye Institute. We also would like to thank the Exome Aggregation Consortium and the groups that provided exome variant data for comparison. A full list of contributing groups can be found at http://exac.broadinstitute.org/about. This Article resulted in part from a successful GeneMatcher match[63].

## Author contributions

G.Y. and V.L.R.-P. with the help of P.L. and C.B. designed and coordinated the study. M.S.Z., G.A.O., S.T., M.A., and M.I. conducted clinical evaluation of family 1 and prepared samples. J.A.C.-M., J.N., V. M.-G., E.F.-N., P.L., and V.L.R.-P. carried out arrays and molecular genetic studies in family 1, C.H.-C. and E.F.T. prepared samples and performed M.L.P.A. G.Y. conducted clinical and genetic evaluation of family 2 and C.H. performed the neuropathology studies. Y.H. prepared samples and additional genetic and clinical analysis for family 2 was performed by S.D., D.D.S., K.C., V.B., and C.B. D.N., R.Z. and J.J.C. provided mice for conditional studies. E.F.-N. and M.C.E. developed mouse lines from founders and phenotyped mice. Mouse histology was carried and assessed by M.I.E. and J.R. with the help of E.F.-N. and M.C.E. who prepared samples and conducted image quantifications. A.L., R.L., and G. H.-B. performed and analyzed D.X.A. M.C.E. and E.F.-N. carried out W.B., immunofluorescence, T.E.M., cell culture and TaqMan expression assays with the help of V.L. R.-P. E.F.-N. performed statistical analysis. G.Y., E.F.-N., M.C.E., and V.L.R.-P. wrote the manuscript and prepared figures adding contributions from all authors.
