## [Peer Review File · Nature Communications]

Reviewers' Comments:

Reviewer #1:

Remarks to the Author:

This manuscript is very interesting since point mutations in a muscle-specific isoform of FXR1 were detected in patients affected by a form of myopathy and reproduced in two knock-in mouse models.

Main criticisms:

1. To discriminate between loss and gain of function induced by the mutations, the authors should produce a mutant preventing splicing generating the exon 15 containing muscle-specific isoform of FXR1.
2. Authors should compare by WB and immunofluorescence expression of FXR1P isoforms in myoblasts and myotubes derived both from patients and from animal models. Indeed, in Fig. 2a-b this is made for one patient, why in Fig. 5b this is made only in muscle extracts.
3. In Fig. 2b an antibody specifically recognizing the 82-84kD isoforms is used (#27-15), but the bands recognized display a band size lower than 75kD. The authors should provide WBs where they detect transfected isoforms of FXR1 with all the antibody they used.
4. Fig. 2b: which antibody is used? According to Dubé et al. (BMC Genomics 2000), 82-84kD isoforms are nuclear in myotubes.
5. Authors should find interactors of the two abnormal FXR1P isoforms by co-immunoprecipitation or by GST pull-down. This will help to identify proteins colocalized with FXR1P in abnormal granules.
6. Are the mutants of FXR1P still able to bind mRNA?
7. The statistical tests used are never indicated in the figure legends; the number of samples used is not precise.

Reviewer #2:

Remarks to the Author:

Estan MG et al NCOMMS 18-10094

The authors describe 5 probands from 2 families carrying mutations in FXR1, more particularly in exon 15 of the muscle-specific isoform.

In the consanguineous family 1 a 4 nucleotide deletion was predicted to replace the last 90 aa of the muscle isoforms. This lead (in the human) to a severe in-utero onset arthrogryposis with profound lack of muscle formation and consequently muscle function leading to joint contractures, bone fractures, swallowing deficit, hypoventilation, and eventually death.

Skeletal muscle was not available from probands of this family, and multi-minicore muscle pathology was not shown. The phenotype looks rather like a muscle aplasia phenotype, potentially resembling more closely to what has been shown in *Fxr1* ko mice (Mientjes et al, 2004).

In family 2 three affected siblings carried a homozygous single nucleotide deletion in exon 15 predicted to cause a frame shift. Parents are said to be non-consanguineous but the father was not available for testing. However, it would be easy to test in the siblings if there are a significant number of shared alleles, particularly around the mutation locus. The phenotype in this family is much milder, compatible with a congenital and slowly progressive multi-minicore myopathy, associated also with some muscle wasting and fatty degeneration over time.

Constitutive inactivation of the *Fxr1* isoforms in the myogenic lineage of mice lead to a neonatal lethal phenotype. However, when introduced into mouse the mutation found in family 1 (severe

human phenotype) caused a mild phenotype with minicore myopathy, associated to a 76.6% reduction of Fxr1 transcript, trapping of the protein in the insoluble fraction, and grossly abnormal subcellular localisation in cytoplasmic globules rather than in parallel to the Z discs.

A second mutation of exon 15 introduced in the mouse (not corresponding to a mutation in the human) lead to an even milder muscle phenotype with evidence of only a few cores.

This manuscript provides a plausible and credible association of mutations in FXR1/Fxr1 to a myopathy, and more specifically for exon 15 mutations (exon 15 being part of the maturation-induced muscle isoform) to a multi-minicore phenotype in mouse and man.

Questions remain why the same mutation, which leads to a very severe phenotype in the human is only associated with a mild phenotype in the mouse. This conundrum will be broken down into several strands of questions:

- 1) Can the authors be sure that in the highly and multiply consanguineous family 1 the FXR1 mutation is the only reason for the severe neonatal phenotype?
- 2) Is there any experimental evidence suggesting that the 4 nucleotide deletion observed in family 1 does or does not disrupt the complete Fxr1 myogenic lineage program (i.e. expression of isoform in myoblasts rather than myotubes)?
- 3) Along the same line: if the 4 nucleotide deletion was introduced into (f.e. iPSC-derived) human myoblasts, would it affect the early and later FXR1 transcriptional program in the same way as in the mouse?

The manuscript defines eventual consequences of the FXR1/Fxr1 mutations from RNAsq-based expression profiling of P 15 mice. As common to such exercise, a fair number of up- or down-regulated gene expressions were observed, and some scholarly interpretation is provided without necessarily providing convincing links with the observed pathology. To name just one example, Cxc110 overexpression in both delACAG and dupA mice was interpreted as evidence of a pro-inflammatory process and in the discussion linked to the bone phenotype in the mice and in the patients of family 1. This is extremely far-fetched and speculative. In the human probands, the degree of bone hypoplasia and fractures is observed in arthrogyrosis of widely different genetic origin and most likely attributable to the in utero akinesia and consequent lack of bone development. In the mouse, no inflammatory phenotype was observed in the primary tissue concerned, skeletal muscle.

On the other hand, no reference is made to a putative function of FXR1 as part of a mRNP protein complex involved in RNA handling (transport, transcriptional handling, translation?), and how this could be linked to Z-line instability.

Finally, in addition to SEPN1 and RYR1, mutations in titin have been associated with (severe) multi-minicore myopathy (Carmignac et al, 2007; Chauveau et al, 2014), and more recently also mutations in DOK7 and MYH7. This discussion needs enlargement, and sequence abnormalities in these genes should specifically be excluded as potential contributors to the phenotype(s) in the probands.

Reviewer #3:

Remarks to the Author:

This is a nice manuscript in which the authors have demonstrated that mutations in FXR1 exon 15, which is exclusively included in transcripts of the gene in muscle cells, can cause multi-minicore myopathy in humans. Complementing the findings of mutations in two human families, the real strength of this study is the authors' use of CRISPR-Cas9 to generate one of the family-specific exon 15 mutations directly in the mouse genome; the resulting homozygous mice persuasively

phenocopied the human disease and established the causal nature of the mutation. The authors went on to show that a different exon 15 mutation generated in one mouse line as a consequence of the semirandom nature of CRISPR-Cas9 NHEJ mutagenesis had a different (milder) phenotypic manifestation, highlighting the diversity of penetrance of exon 15 mutations.

In general, the work appears to me to be technically sound, and I have no suggestions for improvements in that respect. My only suggestion is that the transcriptomic analyses add little to the paper (with respect to mechanistic understanding, which presumably was the intent of the analyses) and could be removed to make the manuscript shorter and more focused.

Reviewer #4:

Remarks to the Author:

Estan et al. present a manuscript in which they describe 5 patients from 2 families with a congenital muscle disease with pathogenic variants in an exon of the FXR1 gene, which is only present in muscle and testis. The authors present the case histories, clinical details, and histopathology of these patients who show multi mini-cores in their muscles. Further, the authors have generated two mouse models with different mutations in FXR1 isoforms e-f using the CRISPR/Cas9 method in the respective muscle specific exons that nicely mirror the human phenotype in many respects including the muscle disease with central cores. In the mice, the authors found a reduced bone density, which might explain the presence of congenital bone fractures in the two index patients. Using isoform specific antibodies the authors were able to demonstrate the wildtype protein to be associated with the Z-bands, while the mutant protein accumulated in granules around the cytoplasm. Overall this is an excellent, carefully executed piece of scientific work with all the necessary controls that describes a new disease, which would be of great interest to the muscle disease community.

[1] It would be helpful to the reader if the authors would present a scheme of the different splicing isoforms (which exons are included and which not) and in which tissues they would be present. Into this scheme they should insert the positions of the respective mutations and from what position onwards a nonsense peptide would be formed and where the stop codon would be located. This would facilitate to understand their reasoning why nonsense mediated decay does occur in one case and not in the other.

[2] I am missing at least an informed discussion of the interesting finding that the dysfunction of the e-f isoform does stimulate the p53 pathway, while the transcript levels of p53 are not increased (measurement on the protein level have not been done). This is a fascinating finding, which the authors do not explore into too much detail (which might be admittedly the subject of another publication). Is there a hint for DNA damage? Is the p53 pathway launched in the absence of DNA-damage? The authors should provide at least some theoretical discussion about what the function of FMX1 at the Z-band could be and what would be the potential link between RNA-transport and the muscle phenotype with multi mini-cores.

[3] Did the author find chromosome instability or decreased condensation of the X-chromosomal chromatin in the muscle of their mouse models (e.g. measured by ATAC-seq or by chromosome analysis in proliferating mouse myoblasts). This should be checked especially given the the fact that FXR1 works in complex with FMR1. I would also suggest to somewhat extend the double-immune-staining experiments of normal muscle, to check whether FXR1 would be found at the Z-band in isolation or in complex with FMR1 and FXR2.

[4] Did the authors also find elevated amounts of Sarcolipin protein at the ER (not only the specific mRNA) in the muscle sections of their transgenic mice? This could be a pathogenetic link to the pathomechanism of RYR mutations and the presence of multi-minicores.

[5] Since the authors have performed WES, please also present a supplemental table of all variants with an allele frequency below 0.1 that have been found in genes already known to cause multi-minicore myopathy.

[6] The authors should mention, how many backcrosses of the CRISPR/Cas9 founder mice into the BL6 line they did for the mice that have finally been analyzed to get an idea, how likely the phenotype in the mouse model could still be caused by off-target effects.

[7] Statistics: the authors mention to have used the Student's t-test to calculate significance levels. How did the authors make sure that their data were normally distributed? If this is not possible (e.g. due to the small sample sizes) please use a non-parametric test (e.g. Man-Whitney-U test).

[8] In figures 3h, 5a, 7d the authors should replace the dynamite plots by dot-plots with each dot representing a measurement.

RESPONSE TO REVIEWERS

We would like to thank all four Reviewers for their positive comments on this manuscript and constructive criticism. Please find our response to each of the comments below.

Reviewer 1

"This manuscript is very interesting since point mutations in a muscle-specific isoform of FXR1 were detected in patients affected by a form of myopathy and reproduced in two knock-in mouse models.

Main criticisms:

1. To discriminate between loss and gain of function induced by the mutations, the authors should produce a mutant preventing splicing generating the exon 15 containing muscle-specific isoform of FXR1."

RE: Compared to wild type mice, dupA mutants synthesize 30% of the normal amount of *FXR1* mRNA and have exon 15 muscle-specific isoforms which are nearly undetectable by WB or immunofluorescence either in muscle tissue or in cell culture differentiated myotubes. Thus, dupA mice are quite close to being null for the muscle-specific isoforms of *FXR1*. We agree with the Reviewer that the suggested new mouse model will represent a total null for the muscle-specific isoforms of FXR1, provided that the new mutation preventing normal splicing of exon15 does not activate any cryptic splice site. However, we feel that the time required to generate, cross and characterize a new mouse mutant will jeopardize the novelty of this manuscript, the main thrust of which is revealing the type of disorder resulting from mutations in the muscle-specific isoforms of *FXR1* and the interesting associated molecular pathology including the assembly of the mutant protein into distinctive granules.

"2. Authors should compare by WB and immunofluorescence expression of FXR1P isoforms in myoblasts and myotubes derived both from patients and from animal models. Indeed, in Fig. 2a-b this is made for one patient, why in Fig. 5b this is made only in muscle extracts."

RE: As requested by the Reviewer we have developed primary myoblast cultures from the mutant mice and have performed WB and immunofluorescence in myoblasts and myotubes from patient (V-4) and both mouse models. These results have been incorporated into the manuscript. Patients from family 2 declined to donate biological samples for these studies.

"3. In Fig. 2b an antibody specifically recognizing the 82-84kD isoforms is used (#27-15), but the bands recognized display a band size lower than 75kD. The authors should provide WBs where they detect transfected isoforms of FXR1 with all the antibody they used."

RE: Antibody #27-15 was kindly provided by Dr. Edouard Khandjian, who is a recognized scientist in the field of FXR1P and Fragile-X proteins. This antibody was generated against an exon15-specific peptide (Dubé M et al, BMC Genetics 2000) which is still present in delACAG and dupA but not in delA isoforms and has been previously used in numerous publications. In Figure 2 there was a slight error in the position of the molecular weight marker. We have exchanged the two WB of Figure 2 for two new immunoblots. As requested we provide in supplemental material WBs for Proteintech and #27-15 FXR1 antibodies showing detection of transfected FXR1P variants by these antibodies (Supplementary Figure 10a-b). Anti-FXR1 antibodies exclusively used in experiments with Myf5-conditionals (#ML13 and #27-17; Supplementary Figure 4c-d) were validated using extracts from *Fxr1* constitutive knockouts. These two antibodies were also kindly provided by Dr. Edouard Khandjian and have been equally used in multiple publications (Mazroui et al Human Mol Genet 2003, Dube et al BMC Genetics 2000).

"4. Fig. 2b: which antibody is used? According to Dubé et al. (BMC Genomics 2000), 82-84kD isoforms are nuclear in myotubes."

RE: In the immunofluorescence experiment of Figure 2b we used Proteintech anti-FXR1P antibody which was raised against the N-terminal region of the protein (1-353 amino acids) and thus recognized all FXR1P isoforms including exon-15 isoforms. Dubé et al used the exon15-specific antibody #27-15 in WB and immunofluorescence and found nuclear staining in myoblasts but not in myotubes. We have not been able to replicate this signal with the Proteintech antibody and also did not see nuclear staining in Hela cells transfected with EGFP-tagged FXR1P variants containing exon 15 (iso e).

"5. Authors should find interactors of the two abnormal FXR1P isoforms by co-immunoprecipitation or by GST pull-down. This will help to identify proteins colocalized with FXR1P in abnormal granules."

RE: Since FXR1P is a key protein of Stress Granules (SGs), after obtaining a positive result in the RNA FISH experiment shown in Figure 8 and Supplementary Figure 9 (please see also response to point 6), we studied co-localization of the mutant FXR1P granules with SG components including TIAR, FXR2P and FMR1P. Dual immunolabelling of FXR1P and RCK was also performed because RCK is a marker for P-bodies which are other type of RNA-protein aggregates. Co-localization of endogenous proteins was assessed in control and patient myoblasts (Figure 9a-b). In addition, since myoblasts contain multiple FXR1P isoforms, we transfected Hela cells with normal or mutant EGFP:FXR1- iso e constructs to specifically ascertain co-localization of exon-15 muscle isoforms with SG components. Remarkably, the mutant iso e variants were capable to assemble in ring-shaped granules in Hela cells (Figure 9c and Supplementary Figure 10c). In this system the wild type iso e worked as a SG protein, but the mutant iso e isoforms once they were assembled into ring-shaped granules did not co-localize with SG components or exhibited weak co-localization (Supplementary Figure 11).

In line with the suggestion of the Reviewer we intend to purify endogenous delACAG-granules from muscle tissue and patient cells and perform both mass

spectrometry and RNA immunoprecipitation to determine the identity of proteins and mRNAs contained by these granules. However, the length of time required to conduct these experiments and to subsequently confirm and validate the resulting candidates compromise the novelty of this manuscript.

"6. Are the mutants of FXR1P still able to bind mRNA?"

RE: To address the Reviewer's question we have conducted fluorescence in situ hybridization in isolated muscle fibres from mouse mutants and control mice and in patient differentiated myotubes using a fluorescently labeled oligodT probe. A similarly labeled scramble probe was used as control. We observed that a fraction of delACAG granules, commonly the bulkier, were positive for oligodT hybridization. In wild type fibres the FISH signal was detected in a periodic pattern resembling Z-lines. Costameric structures have been reported to contain specific mRNAs implicated in muscle contraction which are thought to act as reservoirs of mRNAs required for local de novo protein synthesis (Huot M et al, Mol Biol Cell 2005). These data have been incorporated into the result section and are shown in Figure 8 and in Supplementary Figure 9. No differences were observed in the distribution of mRNA in dupA fibres with respect to controls (data not shown in the manuscript).

"7. The statistical tests used are never indicated in the figure legends; the number of samples used is not precise."

RE: In the previous version of the manuscript we included general statements to minimize repetition. Tests and number of samples are now indicated in each figure.

Reviewer 2

- "The authors describe 5 probands from 2 families carrying mutations in FXR1, more particularly in exon 15 of the muscle-specific isoform. In the consanguineous family 1 a 4 nucleotide deletion was predicted to replace the last 90 aa of the muscle isoforms. This lead (in the human) to a severe in-utero onset arthrogyriposis with profound lack of muscle formation and consequently muscle function leading to joint contractures, bone fractures, swallowing deficit, hypoventilation, and eventually death.

Skeletal muscle was not available from probands of this family, and multi-minicore muscle pathology was not shown. The phenotype looks rather like a muscle aplasia phenotype, potentially resembling more closely to what has been shown in Fxr1 ko mice (Mientjes et al, 2004). In family 2 three affected siblings carried a homozygous single nucleotide deletion in exon 15 predicted to cause a frame shift. Parents are said to be non-consanguineous but the father was not available for testing. However, it would be easy to test in the siblings if there are a significant number of shared alleles, particularly around the mutation locus. The phenotype in this family is much milder, compatible with a congenital and slowly progressive

multi-minicore myopathy, associated also with some muscle wasting and fatty degeneration over time.”

RE: As requested by the Reviewer we performed whole genome SNP-array hybridization in the three patients and a control individual using DNAs purified from lymphoblast cell lines. The three siblings were found to share a region of homozygosity of 5.98 MB comprising *FXR1*. This block of homozygosity is not due to deletion of the paternal chromosome because both the LogR-ratio index obtained in the SNP-arrays and array comparative genomic hybridization (aCGH) did not detect a heterozygous deletion of this region. No other large homozygous blocks were detected in the SNP-arrays of the patients. Thus, it is quite possible that the parents of family 2 share a distant common ancestor (Supplementary Figure 2b-c).

“Constitutive inactivation of the *Fxr1* isoforms in the myogenic lineage of mice lead to a neonatal lethal phenotype. However, when introduced into mouse the mutation found in family 1 (severe human phenotype) caused a mild phenotype with minicore myopathy, associated to a 76.6% reduction of *Fxr1* transcript, trapping of the protein in the insoluble fraction, and grossly abnormal subcellular localisation in cytoplasmic globules rather than in parallel to the Z discs. A second mutation of exon 15 introduced in the mouse (not corresponding to a mutation in the human) lead to an even milder muscle phenotype with evidence of only a few cores. This manuscript provides a plausible and credible association of mutations in *FRX1/Frx1* to a myopathy, and more specifically for exon 15 mutations (exon 15 being part of the maturation-induced muscle isoform) to a multi-minicore phenotype in mouse and man.”

RE: Mice mimicking family1’s mutation (delACAG mutants) express 76.6% of the amount of *Fxr1* mRNA expressed by wild type mice and they were also found to fabricate significant levels of the mutant isoforms by WB. Although delACAG mice survive the perinatal period, we do not think that they have a mild phenotype. Muscle volume of these mutants measured by MRI was found to be decreased by 35%, which is an important reduction for a mouse (please note picture of gastrocnemius in Figure 3d) and we detected a large proportion of highly disorganized fibres at the EM level. In fact, delACAG mice have increased expression of GDF15 (MIC-1) which is a marker of cachexia (Johnen H et al. Tumor-induced anorexia and weight loss are mediated by the TGF-beta superfamily cytokine MIC-1. Nat Med 2007) and reduced expression of the inhibitor of muscle growth myostatin, possibly as a compensatory mechanism.

Nevertheless, it is important to take into account that differences in phenotypic severity between humans and mice are not unusual for many disorders including muscle conditions. The *mdx* mouse for Duchenne muscular dystrophy is a clear example. *Mdx* mice carry a nonsense mutation in exon 23, but only partially mimic the human disease. They have an almost normal life span and mild muscle weakness.

“Questions remain why the same mutation, which leads to a very severe phenotype in the human is only associated with a mild phenotype in the mouse. This conundrum will be broken down into several strands of questions:

1) Can the authors be sure that in the highly and multiply consanguineous family 1 the FXR1 mutation is the only reason for the severe neonatal phenotype?"

RE: Homozygosity mapping is a powerful genetic tool that has been successfully used for the discovery of many genes responsible for human disease. Thus, based on the pedigree (parents are first cousins), we used this approach in combination with whole exome sequencing (WES) in the proband of family 1. We carefully analyzed rare variants placed within homozygous blocks larger than 1 MB with the help of pathogenicity predictor programs, allele frequencies and gene functionality information and found the *FXR1* frameshift variant to be the most likely cause of the disease. The identified *FXR1* mutation segregates with the disease within the pedigree, is comprised within large chromosomal segments of homozygosity in the two sibs tested, is not present in control databases, and relevantly was confirmed to be pathogenic by generating a mouse model carrying exactly the same DNA change. To address the Reviewer's comment we have re-examined the WES data for the presence of rare homozygous and compound heterozygous variants located inside and outside homozygous blocks and detected no further candidates. Nevertheless, as in any other human condition, it could be possible that modifier genes or environmental factors are influencing the phenotype of family 1. Indeed, cases of inter- and even intra-familial phenotypic variability are frequently reported in nearly every single human disorder.

On the other hand, in addition to the possibility of modifier genes, we demonstrate that mice with different mutations in exon15 of *FXR1* have variable muscle phenotypes, with the more severe phenotype being associated with aggregation of the mutant FXR1P isoforms in specific granules. Notably, re-assessment of myofibres from the muscle biopsy of family 2 at the EM level failed to detect delACAG-type granules. On this basis, the phenotypes of families 1 and 2 could be considered equivalent in the mouse to the phenotypes of delACAG and dupA mutants respectively.

Regarding this, Becker muscular dystrophy (BMD, MIM 300376) and Duchene muscular dystrophy (DMD; MIM 310200), are both caused by mutations in the gene encoding dystrophin on chromosome Xp21. BMD is similar to DMD in the distribution of muscle wasting and weakness, which is mainly proximal, but the course is more benign, with wheelchair-dependency after 16 years and some patients having no symptoms until much later in life.

"2) Is there any experimental evidence suggesting that the 4 nucleotide deletion observed in family 1 does or does not disrupt the complete *Fxr1* myogenic lineage program (i.e. expression of isoform in myoblasts rather than myotubes)?"

RE: Proliferating myoblasts express a variety of FXR1P isoforms including P82,84 variants containing exon 15 and smaller variants named P70-80. When myoblasts are differentiated into myotubes, the synthesis of P82,84 is stimulated, while the generation of P70-80 isoforms is repressed.

We have studied the expression of all FXR1P isoforms in myoblasts and myotubes from patient V-4 (homozygous for the delACAG mutation) and corresponding control cells both by WB and immunofluorescence. WB analysis demonstrated that V-4 and control proliferating myoblasts have similar protein levels and expression pattern of P70-80 FXR1P isoforms (iso a,b,c,d), but differ in the expression of P82,84 variants (iso e-f). V-4 myoblasts and V-4 myotubes were both found to generate smaller P82,84 variants, which unlike in the control cells, are present in the non-soluble fraction of protein extracts. Thus, the mutation of family 1 only disrupts the muscle specific variants iso e-f, but not the smaller P70-80 isoforms (Figure 2a, Supplementary Figure 3 and Figure 1 at the end of this letter which shows results from two different antibodies against all FXR1P isoforms (annex)).

We have also checked expression of FXR1P isoforms in myoblasts and myotubes by immunofluorescence (Figure 2b). V-4 and control myoblasts showed similar cytosolic dissemination of FXR1P isoforms, except for the presence of some incipient granular staining in V-4 myoblasts. After differentiation, when only P82,84 variants are expressed, these variants continued having a dispersed cytoplasmic distribution in control myotubes, while the localization of P82,84 in V-4 differentiated cells became restricted exclusively to ring-shaped granules. Size and number of granules in V-4 myotubes were increased with respect to the myoblast stage. Hence, consistent with WBs, the homogenous FXR1P staining in the cytosol of V-4 myoblasts corresponds to the P70-80 isoforms, while the FXR1P granular expression detected in these cells is due to the mutant P82,84 variants (Figure 2b). No differences in the expression pattern of isoforms was found between V-4 and control fibroblasts also indicating that expression of P70-80 isoforms is not affected by the mutation (Supplementary Figure 3).

"3) Along the same line: if the 4 nucleotide deletion was introduced into (f.e. iPSC-derived) human myoblasts, would it affect the early and later FXR1 transcriptional program in the same way as in the mouse?"

We believe that we have answered this question by WB and immunofluorescence analysis of patient-derived primary myoblasts and myobubes of family 1. The suggested iPSC experiment involves introducing the mutation of family 1 by CRISPR-technology in an iPSC cell line, which is already modified by exogenous expression of transcription factors, and differentiating the selected cell line(s) carrying the desired mutation into the myoblast lineage. Due to the high degree of experimental manipulation, we think that this is not going to give a different or more reliable answer than studying human primary myoblasts with the homozygous delACAG mutation (V-4 cells). In addition, given the competition in the field, we are concerned about the length of time required for generating these reagents, which will compromise the novelty of this manuscript.

"The manuscript defines eventual consequences of the FXR1/Fxr1 mutations from RNAsq-based expression profiling of P 15 mice. As common to such exercise, a fair number of up- or down-regulated gene expressions were observed, and some scholarly interpretation is provided without necessarily providing convincing links with the observed pathology. To name just one example, Cxcl10 overexpression in both delACAG and dupA mice was interpreted as evidence of a pro-inflammatory process and in the discussion linked to the bone phenotype in the mice and in the patients of family 1. This is extremely far-fetched and speculative. In the human probands, the degree of bone hypoplasia and fractures is observed in arthrogyrosis of widely different genetic origin and most likely attributable to the in utero akinesia and consequent lack of bone development. In the mouse, no inflammatory phenotype was observed in the primary tissue concerned, skeletal muscle."

RE: As suggested by the Reviewer we have modified the paragraph in the discussion so that it is clear that the bone phenotype of family 1 is likely attributable to the *in utero* akinesia. Notably, a role of FXR1P in regulating pro-inflammatory transcripts in vascular smooth muscle cells has been recently reported and we have also added this reference as it supports our RNAseq results (Herman AB, et al. Cell Rep 2018).

"On the other hand, no reference is made to a putative function of FXR1 as part of a mRNP protein complex involved in RNA handling (transport, transcriptional handling, translation?), and how this could be linked to Z-line instability."

RE:

-We have addressed FXR1P functionality with respect to mRNA handling mentioned by the Reviewer and have performed additional experiments on this subject. We now show that the granules formed by the mutant delACAG protein have the capacity to bind mRNA. Thus, in addition to the loss-of-function of FXR1P resulting from its absence from Z-lines, the assembly of the mutant protein in granules interrupts the normal traffic of mRNAs. We have also conducted experiments to evaluate the functionality of the mutant FXR1P protein in the context of Stress Granules which are cytoplasmic RNA-protein aggregates formed after cellular stress which contain FXR1P and stall mRNA translation. These results have been incorporated into the Results section of the manuscript.

-We have added a paragraph to the Discussion connecting a possible role of FXR1P in repressing translation in costameres with Z-line instability.

-FXR1P was demonstrated to regulate the stability of p21 mRNA and *TERC* via direct interaction (Majamunder et al. Plos Genetics 2016; Davidovic L et al. Plos Genetics 2013). We commented on this and quantified the mRNA levels of both genes in mice by qRT-PCR (Results section).

-Our RNAseq results identified upregulation of sarcolipin (SLN) mRNA in mutant mice, which is a protein directly connected to Ca⁺² homeostasis. We now also show increased protein levels of SLN protein in the mutants. Overexpression of SLN is likely to result in alterations of the Ca⁺² balance between the

sarcoplasmic reticulum and cytosol which in turn could lead to abnormalities in the contraction of fibres and stability of Z-lines.

"Finally, in addition to *SEPN1* and *RYR1*, mutations in titin have been associated with (severe) multi-minicore myopathy (Carmignac et al, 2007; Chauveau et al, 2014), and more recently also mutations in *DOK7* and *MYH7*. This discussion needs enlargement, and sequence abnormalities in these genes should specifically be excluded as potential contributors to the phenotype(s) in the probands."

RE:

-As suggested by the Reviewer, we have expanded the list of genes associated with multi-minicore myopathy in the corresponding paragraph of the Discussion. Interestingly, the two *TTN* papers mentioned by the Reviewer are also an example of variability in the severity of the phenotype between patients with mutations in the same gene. All patients from these families had at least one *TTN* truncating mutation. In the report by Carmignac et al. (Ann Neurol 2007) all patients from two families died at ages between 8-19.5 years, while in the report by Chauveau et al. (Hum Mol Genet 2014) all patients from 4 families were between 39-55 years of age at the time of echocardiogram and ECG.

-We have included a supplementary table (supplementary table 2) with WES variants detected in mini-core genes in both families with $MAF < 0.1$ (referee 4). For family 1, variants in recessive genes that were classed as VUS (variant of unknown significance) or with no definition in ClinVar (NCBI) and were particularly rare ($MAF < 0.1\%$) were analyzed by Sanger sequencing in all members of the family (proband, V4 and both parents). Only one variant in *TTN* was also present in V-4 in the heterozygous state and in the unaffected mother. In addition, although not concordant with the pedigree structure, we studied segregation of a dominant variant in *CCDC78* (W410S) in family 1. A single dominant mutation has been reported in this gene in a family with dominant inheritance. Segregation analysis showed the W410S variant in the heterozygous state also in V-4, but the variant involves a non-conservative amino acid, is multiallelic with one of the alleles resulting in W410Stop ($MAF = 0.00003$), and relevantly, was also present in the unaffected mother. Furthermore, the *CCDC78* variant (*rs893572894*) is classed as synonymous when a different reference sequence is used (dbSNP, NCBI).

Reviewer 3

"This is a nice manuscript in which the authors have demonstrated that mutations in *FXR1* exon 15, which is exclusively included in transcripts of the gene in muscle cells, can cause multi-minicore myopathy in humans. Complementing the findings of mutations in two human families, the real strength of this study is the authors' use of CRISPR-Cas9 to generate one of the family-specific exon 15 mutations directly in the mouse genome; the resulting homozygous mice persuasively phenocopied the human disease and established the causal nature of the mutation. The authors went on to show that a different exon 15 mutation generated in one mouse line as a consequence of the semirandom nature of CRISPR-Cas9 NHEJ mutagenesis had a different

(milder) phenotypic manifestation, highlighting the diversity of penetrance of exon 15 mutations.

In general, the work appears to me to be technically sound, and I have no suggestions for improvements in that respect. My only suggestion is that the transcriptomic analyses add little to the paper (with respect to mechanistic understanding, which presumably was the intent of the analyses) and could be removed to make the manuscript shorter and more focused."

RE: We thank the Reviewer for the positive comments on our work.

If there is no objection, we would prefer to leave the RNAseq information in the manuscript because it illustrates pathways and genes altered by the muscle-specific delACAG mutation *in vivo*. This is of interest given that among other functions FXR1P has been shown to be involved in mRNA stability. We have deposited the results of this experiment in the NCBI SRA database and we believe it will be of use not only to us, but also to other scientists working in the Fragile X family of proteins or muscle diseases.

Reviewer 4

"Estan et al. present a manuscript in which they describe 5 patients from 2 families with a congenital muscle disease with pathogenic variants in an exon of the FXR1 gene, which is only present in muscle and testis. The authors present the case histories, clinical details, and histopathology of these patients who show multi mini-cores in their muscles. Further, the authors have generated two mouse models with different mutations in FXR1 isoforms e-f using the CRISPR/Cas9 method in the respective muscle specific exons that nicely mirror the human phenotype in many respects including the muscle disease with central cores. In the mice, the authors found a reduced bone density, which might explain the presence of congenital bone fractures in the two index patients. Using isoform specific antibodies the authors were able to demonstrate the wildtype protein to be associated with the Z-bands, while the mutant protein accumulated in granules around the cytoplasm. Overall this is an excellent, carefully executed piece of scientific work with all the necessary controls that describes a new disease, which would be of great interest to the muscle disease community."

[1] It would be helpful to the reader if the authors would present a scheme of the different splicing isoforms (which exons are included and which not) and in which tissues they would be present. Into this scheme they should insert the positions of the respective mutations and from what position onwards a nonsense peptide would be formed and where the stop codon would be located. This would facilitate to understand their reasoning why nonsense mediated decay does occur in one case and not in the other."

RE: As suggested by the Reviewer, we have amended Supplementary Figure 1 to indicate the position of mutations and stop codons to delineate which mutations activate NMD. To avoid duplication of published information we show the two muscle isoforms which are affected by the mutation and refer to the literature for the exon composition of the remaining FXR1P isoforms (Dubé M et al. BMC Genet 2000 and Kirkpatrick LL et al. Genomics 1999).

[2] I am missing at least an informed discussion of the interesting finding that the dysfunction of the e-f isoform does stimulate the p53 pathway, while the transcript levels of p53 are not increased (measurement on the protein level have not been done). This is a fascinating finding, which the authors do not explore into too much detail (which might be admittedly the subject of another publication). Is there a hint for DNA damage? Is the p53 pathway launched in the absence of DNA-damage? The authors should provide at least some theoretical discussion about what the function of FMX1 at the Z-band could be and what would be the potential link between RNA-transport and the muscle phenotype with multi mini-cores.

a) RE (p53 and DNA damage):

We measured p53 at the protein level by immunoblotting, but similar to the mRNA levels, we found no reliable differences between normal and mutant mice. Since p73 and p63 share targets with p53, the increased expression of these genes is likely to be responsible for the activation of the p53-pathway detected in the RNA-seq. We have investigated DNA damage by testing foci accumulation of γ -H2AX and TP53BP1 in nuclei by immunofluorescence in V-4 myotubes without convincing evidence. Additionally, we have checked whether the delACAG mutation could induce a DNA damage response by stimulating mobilization of LINE-1 or endogenous retrovirus-like mobile elements (intracisternal A particles-IAPs). FXR1P has been suggested to be involved in the control of these elements in Stress Granules (Goodier JL et al. *Mol Cell Biol* 2007; Kim DY et al. *Plos Pathog* 2016). We studied expression of these mobile elements in muscle of normal and mutant mice by qRT-PCR, but again obtained no positive result. Nevertheless, we want to investigate the possibility of DNA damage in these mice further using additional reagents and approaches. As the Reviewer indicates this could be the starting point of another publication.

-Goodier JL et al. LINE-1 ORF1 protein localizes in Stress Granules with other RNA-binding proteins, including components of RNA interference RNA-Induced Silencing Complex. *Mol Cell Biol*, 2007.

-Kim DY et al. New World and Old World alphaviruses have evolved to exploit different components of Stress Granules, FXR and G3BP proteins, for Assembly of Viral replication complexes. *Plos Pathog*, 2016.

b) RE (p53 pathway activation in the absence of DNA damage):

We believe that p73/p63 overexpression can explain the activation of the p53 pathway in the absence of DNA damage. These two p53-homologs are known to be more involved in development than p53. Strong activation of p63 has already been specifically demonstrated in muscle atrophy and levels of p73 have been shown to progress from undetectable in proliferating C2C12 myoblasts to overexpressed after differentiation (von Grabowiecki Y et al. *Elife* 2016; Fontemaggi et al. *MCB*, 2001). Thus, both p63 and p73 have functions beyond the DNA damage response related to muscle development, although their exact roles remain to be elucidated.

-von Grabowiecki Y et al., Transcriptional activator TAp63 is upregulated in muscular atrophy during ALS and induces the pro-atrophic ubiquitin ligase Trim63. *Elife*, 2016.

-Fontemaggi G et al. The Transcriptional Repressor ZEB Regulates p73 Expression at the Crossroad between Proliferation and Differentiation. *Mol Cell Biol*, 2001.

c) RE (theoretical discussion about FXR1P function at Z-band and link with RNA and multi mini-core phenotype):

-We have added the following paragraph in the discussion: "It has been suggested that similar to the role of FMR1P in neurons, FXR1P could act in muscle maintaining specific mRNAs in a repressed state in costamere structures until they become required to be translated for de novo synthesis of proteins (Huot ME et al. *Mol Biol Cell*. 2005). Consistent with this hypothesis we observed that FXR1P-iso e can be recruited into SGs, which are structures involved in translation repression. Hence, hypothetically, de-regulation of translation of specific mRNAs involved in Z-line organization could underlie the minicore phenotypes resulting from P82,84 mutations".

- We show that delACAG mutants, which have a more severe phenotype than dupA mice, can sequester mRNA molecules in granules and thus can also impair mRNA traffic and turnover. This has been added to the Results and subsequently commented in the Discussion.

-Please also see our response to Reviewer 2 (comment 6).

"[3] Did the author find chromosome instability or decreased condensation of the X-chromosomal chromatin in the muscle of their mouse models (e.g. measured by ATAC-seq or by chromosome analysis in proliferating mouse myoblasts). This should be checked especially given the the fact that FXR1 works in complex with FMR1. I would also suggest to somewhat extend the double-immune-staining experiments of normal muscle, to check whether FXR1 would be found at the Z-band in isolation or in complex with FMR1 and FXR2".

RE: Instability of the X chromosome and chromatin condensation changes at the *FMR1* locus in Fragile X syndrome are due to expansion of CGG repeats which induce methylation and silencing of the *FMR1* promoter. Accordingly, inactivation of the FMR1P protein is secondary to CGG triplet expansion. Nevertheless, we have checked FMR1P expression by WB in muscle extracts from wild type and delACAG mice using previously published validated antibodies (DSHB; 7G1-1). We have observed a slight increase in FMR1P protein levels and thus, we think that loss-of-function changes in FMR1P are not expected in these mice. Please note that FMR1P expression is considerably low in muscle (Bakker CE et al. *Exp Cell Res* 2000).

"[4] Did the authors also find elevated amounts of Sarcolipin protein at the ER (not only the specific mRNA) in the muscle sections of their transgenic mice? This could be a pathogenetic link to the pathomechanism of RYR mutations and the presence of multi-minicores."

RE: We thank the Reviewer for this suggestion. To answer this question we obtained an antibody against sarcolipin (SLN) from Dr. Robert Bloch who is an expert on SLN studies. This antibody has been previously published and shown to detect both transfected and endogenous SLN protein (Desmond PF et al. J Biol Chem 2017). We have analyzed protein levels of SLN in the soleus and gastrocnemius of normal and delACAG mice and observed a considerable increase in the expression of this protein in mutant mice, thus confirming the mRNA results. This has been incorporated into the manuscript (Figure 10f and Supplementary Figure 12b-c).

"[5] Since the authors have performed WES, please also present a supplemental table of all variants with an allele frequency below 0.1 that have been found in genes already known to cause multi-minicore myopathy."

RE: As suggested by the Reviewer we have added supplementary table I containing this information.

"[6] The authors should mention, how many backcrosses of the CRISPR/Cas9 founder mice into the BL6 line they did for the mice that have finally been analyzed to get an idea, how likely the phenotype in the mouse model could still be caused by off-target effects."

RE: Two founders were used for the delACAG mutation and one founder for the dupA variant. All the three mice were generated with the same CRISPR RNA guides. We established three mouse lines, one from each founder, which were expanded and maintained independently by crossing heterozygous males of each line to wild type C57BL/6 females. Results from both delACAG mouse lines were identical. Up to now we have used homozygotes and control wild type mice offspring of F2 to F5 heterozygous crosses without finding any difference in the phenotypes. This has been inserted in material and methods of the manuscript.

"[7] Statistics: the authors mention to have used the Student's t-test to calculate significance levels. How did the authors make sure that their data were normally distributed? If this is not possible (e.g. due to the small sample sizes) please use a non-parametric test (e.g. Man-Whitney-U test."

RE: Following the advice of the Reviewer we have re-assessed statistics and have used corresponding tests to determine distribution normally and homoscedasticity of variance. When assumptions of normality and homoscedasticity of variance were not met, we used non-parametric Kruskal-Wallis with Dunn's multiple comparison test as post hoc. In each figure we have indicated the type of test used and have specified the exact number of samples (as also requested by Reviewer1).

"[8] In figures 3h, 5a, 7d the authors should replace the dynamite plots by dot-plots with each dot representing a measurement."

RE: This has been changed according to the suggestions of the Reviewer.

ANNEX

Figure 1

Figure 1. Expression of FXR1P isoforms in patient and control primary myoblasts and myotubes. Representative immunoblots performed with two different FXR1P antibodies capable to recognize all FXR1P isoforms. Cell extracts (soluble (SF) and insoluble (IF) fractions) from proliferating myoblasts and differentiated myotubes derived from V-4 and a normal control were analyzed. For clarity reasons blots on the top have been split in two halves underneath, so that we could label each isoform in control and V-4 lanes. Isoforms are labelled according to their molecular weight as in Davidovic L et al. JMG, 2014.

V-4 and control cells show similar expression of P70-80 variants (iso a, b, c, d), but differ in the molecular size and solubility of P82,84 variants (iso e-f). Top panel (a): Proteintech (13194-1-AP) polyclonal anti-FXR1P antibody. Lower panel (b): Millipore (clone 6BG10) monoclonal anti-

FXR1P antibody. Please note that Davidovic L et al. reported similar pattern of FXR1P variants in human primary myoblasts with a third different antibody (Davidovic L et al. JMG, 2014). Thus, at least three different antibodies reproduce the same pattern of FXR1P isoforms in extracts from human myoblasts and myotubes. Asterisks in panel a denote a non-specific band which in the control cells is hidden by the P82,84 variants. We have validated both Proteintech and Millipore antibodies in cells transfected with FXR1P variants. The corresponding data for the Proteintech antibody is in Supplementary Figure 10a-b. The validating blot for the Millipore antibody is not in the manuscript because we have not included results from this antibody in the revised version. Nevertheless, we can provide it, if required. Both antibodies have been previously reported.

Reviewers' Comments:

Reviewer #1:

Remarks to the Author:

The authors answered to my criticisms

Reviewer #4:

Remarks to the Author:

The authors have addressed all my suggestions and concerns adequately. Thank you very much for taking your time to do so.

Reviewer #5:

Remarks to the Author:

The authors have adequately addressed the concerns of referee 2.

However, there are some issues that could be cleaned up in reference to the muscle pathology illustrated in Figure 1. Because the authors chose to classify this congenital myopathy as "multiminicore myopathy", it is incumbent to provide strong supportive evidence for this. Skeletal muscle from only one of the 5 individuals with FXR1P exon 15 mutations was evaluated. We don't know how representative this muscle biopsy is for humans with this genetic disease. The word description of the biopsy on pages 7 and 8 of the manuscript is poorly illustrated in Figure 1. There certainly is type 1 fiber predominance (triceps muscle is typically about 50% type 1 fibers), but otherwise the images show no fatty infiltration, no increased variability in muscle fiber size, and no significant increase in internalized nuclei. In a diagnostic report, this muscle biopsy H&E image would be considered "mild nonspecific changes" or "no diagnostic abnormality". The size scale in Figure 1e is not accurate, as it indicates the muscle fibers to be around 300 to 400 microns in diameter. The diameters are more likely in the 50-80 micron range. There is no scale bar for the plastic section toluidine blue image or the electron micrograph. The toluidine blue-stained plastic section image could just as easily be artifacts as minicores. If cores are truly present and identifiable by light microscopy, a more widely accepted way to illustrate them would be to show an oxidative enzyme histochemical stain (NADH, SDH, COX, or combined SDH-COX). Perhaps a better solution to illustrating minicores would be to replace the toluidine blue image with an electron micrograph of muscle in longitudinal orientation. Cores are much easier to appreciate in longitudinal orientation than in cross sectional orientation. Adding a longitudinal orientation electron micrograph and keeping the cross sectional image already part of Figure 1 may make a more convincing case that the biopsy has multiple small cores. Another mismatch between the figure and the text is the description on page 8 that "most lesions were limited to one or two sarcomeres in size" when the electron micrograph shows disruptions of a size that is equivalent to perhaps 10 or more adjacent sarcomeres. Maybe the authors mean the cores are typically one to two sarcomeres in length, which they have not illustrated.

The MRI images in Figure 1 are poor quality. Perhaps they should be omitted, because they contribute very little to the message of the paper.

The skeletal muscle minicores are well illustrated in the delACAG and dupA mice.

On page 23 (lines 548-551) of the discussion, the authors use the term "symptoms" when they are describing pathologic features of the muscle. Muscle atrophy, multicore lesions, central nuclei, and type 1 fiber predominance are not symptoms.

Since the same FXR1P isoform expressed in skeletal muscle is said to be the only isoform

expressed in cardiac muscle, why is there no apparent myocardial pathology or dysfunction? Perhaps this should be addressed in the discussion.

Nowhere in the manuscript do the authors use the term arthrogryposis. The patient illustrated in Figure 1b appears to be classic arthrogryposis multiplex congenita. It may be important to include this terminology somewhere in the paper, perhaps in the discussion where the authors have already briefly mentioned "decreased fetal movement".

Nature Communications
November 6th, 2018
RE: NCOMMS-18-10094A

RESPONSE TO REVIEWERS

We thank all previous Reviewers and the new Reviewer 5 for their constructive comments and suggestions which have helped to improve the manuscript.

Please find our response to each of the new comments below.

Reviewer 1

"The authors answered to my criticisms".

Reviewer 4

"The authors have addressed all my suggestions and concerns adequately. Thank you very much for taking your time to do so."

Reviewer 5

"The authors have adequately addressed the concerns of referee 2.

However, there are some issues that could be cleaned up in reference to the muscle pathology illustrated in Figure 1. Because the authors chose to classify this congenital myopathy as "multi-minicore myopathy", it is incumbent to provide strong supportive evidence for this. Skeletal muscle from only one of the 5 individuals with FXR1P exon 15 mutations was evaluated. We don't know how representative this muscle biopsy is for humans with this genetic disease. The word description of the biopsy on pages 7 and 8 of the manuscript is poorly illustrated in Figure 1. There certainly is type 1 fiber predominance (triceps muscle is typically about 50% type 1 fibers), but otherwise the images show no fatty infiltration, no increased variability in muscle fiber size, and no significant increase in internalized nuclei. In a diagnostic report, this muscle biopsy H&E image would be considered "mild nonspecific changes" or "no diagnostic abnormality". The size scale in Figure 1e is not accurate, as it indicates the muscle fibers to be around 300 to 400 microns in diameter. The diameters are more likely in the 50-80 micron range. There is no scale bar for the plastic section toluidine blue image or the electron micrograph. The toluidine blue-stained plastic section image could just as easily be artifacts as minicores. If cores are truly present and identifiable by light microscopy, a more widely accepted way to illustrate them would be to show an oxidative enzyme histochemical stain (NADH, SDH, COX, or combined SDH-COX). Perhaps a better solution to illustrating minicores would be to replace the toluidine blue image with an electron micrograph of muscle in longitudinal orientation. Cores are much easier to appreciate in longitudinal orientation than in cross sectional orientation. Adding a longitudinal orientation electron micrograph and keeping the cross sectional image already part of Figure 1 may make a more convincing case that the biopsy has multiple small cores."

RE: To address the Reviewer's comment we have revised Figure 1 and have incorporated new images including: a new H-E micrograph that illustrates more clearly the occurrence of nuclear internalization in muscle fibers (Fig. 1e); ii) a Masson's trichrome photomicrograph to demonstrate fatty infiltration (Fig. 1f); and a better quality image for ATPase-pH 4.3 (Fig. 1g). Corresponding scale bars are now present in each image. We have removed the statement about increased variability in muscle fiber size, since as indicated by the reviewer, it is a mild feature. In addition, as also suggested by the Reviewer, we have replaced the toluidine blue image by two electron micrographs in longitudinal orientation which prove the presence of multiple small cores in the muscle fibers of the patient. Unfortunately, as we indicated in the previous revision, patients from family 2 declined to donate additional biological samples and we have no tissue from patients of family 1. However, we believe that the exon 15 mouse models described in this manuscript provide support for the human pathology associated with mutations in *FXR1*-exon15 since they replicate most pathological features of the muscle biopsy of family 2 including minicores.

- "Another mismatch between the figure and the text is the description on page 8 that "most lesions were limited to one or two sarcomeres in size" when the electron micrograph shows disruptions of a size that is equivalent to perhaps 10 or more adjacent sarcomeres. Maybe the authors mean the cores are typically one to two sarcomeres in length, which they have not illustrated."

RE: The Reviewer is right and we have changed the statement accordingly. The new longitudinal electron micrographs included in figure 1h prove the presence of multiple small cores, most of which are one to two sarcomeres in length.

- "The MRI images in Figure 1 are poor quality. Perhaps they should be omitted, because they contribute very little to the message of the paper."

RE: MRI images have been omitted from figure 1.

- "The skeletal muscle minicores are well illustrated in the delACAG and dupA mice."

RE: We thank the Reviewer for this comment.

- "On page 23 (lines 548-551) of the discussion, the authors use the term "symptoms" when they are describing pathologic features of the muscle. Muscle atrophy, multicore lesions, central nuclei, and type 1 fiber predominance are not symptoms."

RE: We agree with the Reviewer and have replaced the term "symptoms" by "pathologic features".

- "Since the same FXR1P isoform expressed in skeletal muscle is said to be the only isoform expressed in cardiac muscle, why is there no apparent myocardial pathology or dysfunction? Perhaps this should be addressed in the discussion."

RE: We have inserted the following paragraph in the Discussion:

“Considering cardiac muscle, the proband of family 1 was found to have episodes of tachycardia, but no apparent myocardial pathology or dysfunction was detected in patients of family 2. However, since P82,84 are the main FXR1P isoforms in cardiac muscle, a pathogenic effect of *FXR1*-exon15 mutations in this tissue cannot be discarded. Future histopathological and functional analysis of the heart musculature of delACAG and dupA mice will inform of this possibility”.

-“Nowhere in the manuscript do the authors use the term arthrogryposis. The patient illustrated in Figure 1b appears to be classic arthrogryposis multiplex congenita. It may be important to include this terminology somewhere in the paper, perhaps in the discussion where the authors have already briefly mentioned “decreased fetal movement”.

RE: Following the suggestion of the Reviewer we have mentioned the term arthrogryposis in the last paragraph of the discussion.

Reviewers' Comments:

Reviewer #5:

Remarks to the Author:

The revised manuscript adequately addresses most issues I raised earlier. One remaining problem is the illustration of internalized nuclei in the patient's muscle biopsy (Figure 1e). In order to clearly see the nuclei, the full page figure supplied by the authors must be magnified to about 200%. In print form, this figure will likely not be full page and no one will be able to see the nuclei. Since there are many muscle fibers with internal nuclei in the field that was photographed, simply replace the current image for 1e with one taken at a higher magnification. This will likely require re-photographing the glass slide. Simply cropping the image and electronically increasing the size of the remaining image is likely to result in pixelation.

The size bars and lettering used for labeling the size bars may be improved by making them white for figures 1f, 1g, and 1h. Changing the color of the size bar and lettering from black to white may improve the readability in several other figures.

Nature Communications
December 20th, 2018
RE: NCOMMS-18-10094B

RESPONSE TO REVIEWERS

We thank again Reviewer 5 for constructive suggestions to improve the presentation of our manuscript.

Reviewer 5

"The revised manuscript adequately addresses most issues I raised earlier. One remaining problem is the illustration of internalized nuclei in the patient's muscle biopsy (Figure 1e). In order to clearly see the nuclei, the full page figure supplied by the authors must be magnified to about 200%. In print form, this figure will likely not be full page and no one will be able to see the nuclei. Since there are many muscle fibers with internal nuclei in the field that was photographed, simply replace the current image for 1e with one taken at a higher magnification. This will likely require re-photographing the glass slide. Simply cropping the image and electronically increasing the size of the remaining image is likely to result in pixelation.

RE: Following the advice of the Reviewer we have replaced the image corresponding to Fig.1e with other taken at a higher magnification re-photographed from the glass slide.

"The size bars and lettering used for labeling the size bars may be improved by making them white for figures 1f, 1g, and 1h. Changing the color of the size bar and lettering from black to white may improve the readability in several other figures."

RE: As suggested by the Reviewer we have changed size bars to white in Figures 1f, 1g, 1h and others. Following editorial instructions lettering for all size bars is now in Figure legends.